# MLP-KAN: Unifying Deep Representation and Function Learning

## Abstract

Recent advancements in both representation learning and function learning have demonstrated substantial promise across diverse domains of artificial intelligence. However, the effective integration of these paradigms poses a significant challenge, particularly in cases where users must manually decide whether to apply a representation learning or function learning model based on dataset characteristics. To address this issue, we introduce MLP-KAN, a unified method designed to eliminate the need for manual model selection. By integrating Multi-Layer Perceptrons (MLPs) for representation learning and Kolmogorov-Arnold Networks (KANs) for function learning within a Mixture-of-Experts (MoE) architecture, MLP-KAN dynamically adapts to the specific characteristics of the task at hand, ensuring optimal performance. Embedded within a transformer-based framework, our work achieves remarkable results on four widely-used datasets across diverse domains. Extensive experimental evaluation demonstrates its superior versatility, delivering competitive performance across both deep representation and function learning tasks. These findings highlight the potential of MLP-KAN to simplify the model selection process, offering a comprehensive, adaptable solution across various domains.

## 1 Introduction

In recent years, deep learning has evolved from early neural network concepts to sophisticated architectures, such as transformer networks (Vaswani, 2017), driven by advancements in computational resources and the availability of large datasets, thereby achieving remarkable performance across diverse applications. Alongside these technological breakthroughs, representation learning (OpenAI, 2023a; Anthropic, 2024; OpenAI, 2023b; Touvron et al., 2023) and function learning (Narayan et al., 1996; Zhang et al., 2022; Wu et al., 2005) have gained prominence and been extensively explored in various research and application tasks. At the same time, the focus of function learning research has shifted from simple function fitting to deep learning (Cuomo et al., 2022; Cai et al., 2021), which excels in tasks requiring precise function approximation and has seen new advancements, particularly in its applicability to univariate function tasks. The key difference between representation learning and function learning lies in their objectives: representation learning aims to extract features from data to understand its underlying structure (Bengio et al., 2013), while function learning focuses on creating direct mappings between inputs and outputs, making it more suited for tasks requiring precise functional relationships (Zupan et al., 1997).

In this paper, we introduce MLP-KAN, a novel framework that unifies two distinct learning approaches into a cohesive system, utilizing the Mixture of Experts (MoE) methodology Jiang et al. (2023).MLP-KAN integrates Kolmogorov-Arnold Networks (KAN) (Liu et al., 2024) and Multi-Layer Perceptrons (MLP) (Rumelhart et al., 1986) , each tailored for specific learning tasks.KANs replace static weights with learnable spline functions, enabling fine-grained interpolation and scalable precision Ta (2024); Somvanshi et al. (2024). These properties make KANs ideal for tasks like symbolic regression, solving partial differential equations (PDEs), and fitting scientific data Liu et al. (2024). In contrast,In contrast, MLPs excel at feature learning by extracting high-level, abstract representations from high-dimensional data (Tashakkori et al., 2024). By employing fixed activation functions and dense weight matrices, MLPs are well-suited for capturing global patterns in applications such as image classification and language modeling.

Figure 1: The comparison between the MLP, KAN, and our proposed MLP-KAN. In the domains of Computer Vision and Natural Language Processing, the goal is to achieve the highest accuracy possible. In contrast, for the Symbolic Formula Representation task, the objective is to minimize the root mean square error (RMSE). The numbers are the average values of the experimental results. MLP-KAN effectively combines the strengths of both, ensuring strong performance in representation and function learning, and eliminating the need for task-specific model selection.

Within the architecture of MLP-KAN, MLP function as representation expert, while KAN is designated as function expert. The MoE mechanism efficiently routes inputs to the appropriate expert, significantly enhancing performance across a diverse range of tasks. MLP-KAN was developed to address the challenge of choosing between representation learning and function learning models for diverse datasets. By integrating MLPs and KANs within a mixture-of-experts framework, this architecture dynamically adapts to the task, as shown in Figure 1, ensuring optimal performance without requiring manual model selection. The main challenge in our method is effectively integrating MLPs and KANs, ensuring the right model is selected for each task without compromising performance. Additionally, aligning the distinct training requirements of representation and function learning, while maintaining efficiency across diverse datasets, presents a significant challenge.

To address the integration of MLPs and KANs within the MoE framework, we utilized a soft MoE approach. This method enables dynamic and flexible routing between MLPs for representation learning and KANs for function learning. By incorporating this MoE system within a transformer framework, the model seamlessly adapts to the task, performing either representation or function learning while maintaining efficiency across diverse datasets.

The main contributions of this work are as follows:

- We present MLP-KAN, a unified framework that synergizes MLP for representation learning with KAN for function learning. This novel architecture leverages a MoE mechanism to dynamically route tasks between representation and function experts, addressing the challenge of selecting the appropriate learning paradigm for diverse datasets.

- We propose a flexible and versatile model by integrating MLP-KAN within the transformer architecture, enabling efficient performance across both representation and function learning tasks. This integration enhances model capability and improves performance across a broad range of tasks, including computer vision, natural language processing, and symbolic formula representation.

- We perform extensive experimental evaluations, demonstrating that MLP-KAN consistently outperforms or matches state-of-the-art models such as MLP and KAN on widely recognized benchmarks, including computer vision,nature language processing, and functional dataset. Our approach achieves superior accuracy in representation learning tasks and lower RMSE in function learning tasks, underscoring its universal applicability across diverse domains.

## 2 RELATED WORK

**Deep Representation Learning.** Deep representation learning has gained significant attention due to its ability to automatically discover hierarchical feature representations from raw data (Butepage et al., 2017; Zhong et al., 2016; Long et al., 2018), outperforming traditional hand-crafted feature extraction techniques. The introduction of deep learning methods, such as MLP based convolutional neural networks (Li et al., 2021) and recurrent neural networks, enabled breakthroughs in

areas like image recognition (Zoph et al., 2018; He et al., 2016), object detection (Zhao et al., 2019; Yu et al., 2016; Liu et al., 2020), and natural language processing (Chowdhary & Chowdhary, 2020; Khurana et al., 2023) by capturing more abstract and high-level features. Recent advancements in deep architectures, including transformer-based models (Gillioz et al., 2020), have further pushed the boundaries of representation learning, proving highly effective across diverse domains. For example, generative AI, such as large language models (LLMs) (Yao et al., 2024; Zhao et al., 2023), has garnered significant attention for its ability to generate coherent, contextually relevant text and learn deep representations from vast amounts of unstructured data. LLMs like GPT-4o (OpenAI, 2024) and LLaMA (Touvron et al., 2023) utilize MLP based transformer architectures, which excel at capturing long-range dependencies in sequential data, allowing them to perform tasks such as text generation, summarization, and translation with remarkable accuracy. Beyond natural language processing, LLMs have also influenced other fields, including code generation (Chung et al., 2024; Li et al., 2022), medical diagnosis (Kononenko, 2001; Amato et al., 2013), and drug discovery (Drews, 2000; Sliwoski et al., 2014), by leveraging their deep learning capabilities to model complex relationships in data. These advancements highlight the growing importance of deep representation learning in not only understanding and generating human-like text but also in solving a wide range of interdisciplinary challenges (Newell et al., 2001). In these models, MLP play a crucial role as fundamental building blocks, serving as dense layers that transform and learn high-dimensional representations by mapping inputs to deeper abstract features (Donoho et al., 2000).

**Deep Function Learning.** Deep function learning focuses on capturing complex mathematical relationships and patterns within data, particularly in scientific and engineering domains (Sarker, 2021; Shen, 2018; Karpatne et al., 2017). Techniques such as Physics-Informed Neural Networks (PINNs) (Raissi et al., 2019) have emerged as powerful tools for solving partial differential equations (PDEs) (Evans, 2022) by embedding physical laws into neural network architectures, allowing for accurate modeling of phenomena governed by underlying physical principles (Raissi et al., 2019; Cuomo et al., 2022). Beyond traditional neural networks, deep function learning leverages over-parameterized models, which enable the precise interpolation of data, even in the presence of noise, enhancing both generalization and optimization performance (Karniadakis et al., 2021; Advani et al., 2020; Chen et al., 2022). Recent advancements have demonstrated the potential of these methods for tasks such as surrogate modeling (Razavi et al., 2012), sensitivity analysis (Christopher Frey & Patil, 2002; Lenhart et al., 2002), and discovery of new scientific relationships (Wren et al., 2004; Klahr & Simon, 1999). KAN are highly effective for function learning due to their ability to capture complex non-linear relationships through learnable spline-based univariate functions, offering superior approximation capabilities and scaling compared to traditional MLP (Yu et al., 2024; Liu et al., 2024; Zhang, 2024; Vaca-Rubio et al., 2024).

## 3 PRELIMINARY

Table 1: Comparison between MLP and KAN.

| Feature | MLPs | KANs |
|---|---|---|
| **Activation Functions** | Fixed functions (e.g., ReLU, SiLU) | $\varphi(x) = \sum_{i=1}^{k} c_i B_i(x)$ |
| **Weight Structure** | Scalar weights | Spline-based weights $\varphi(x)$ |
| **Layer Architecture** | Standard fixed depth | $\Phi_q \left( \sum_{p=1}^{n} \varphi_{q,p}(x_p) \right)$ |
| **Error Scaling** | Limited by dimensionality | $\|f - (KAN)\|_{C^m} \leq CG^{-k-1+m}$ |
| **Scaling Law** | $\ell \propto N^{-\alpha}$ with lower $\alpha$ | $\ell \propto N^{-\alpha}$ with higher $\alpha = 4$ |
| **Expressiveness** | Suited for general representation learning | Suited for functional learning |

KAN are inspired by the Kolmogorov-Arnold Representation Theorem (Liu et al., 2024), which asserts that any multivariate continuous function $f(x)$ can be decomposed into a sum of univariate functions. This is formally stated as:

$$f(x) = \sum_{q=1}^{2n+1} \Phi_q \left( \sum_{p=1}^{n} \varphi_{q,p}(x_p) \right) \tag{1}$$

where $\varphi_{q,p}(x_p)$ and $\Phi_q$ are univariate functions, summing over $q$ and $p$. Unlike traditional Multi-Layer Perceptrons (MLPs), which use fixed activation functions at each neuron, KANs introduce learnable univariate activation functions on the edges between layers (Vaca-Rubio et al., 2024; Aghaei, 2024). Each weight in KANs is replaced by a learnable spline function:

$$\varphi(x) = \sum_{i=1}^{k} c_i B_i(x) \tag{2}$$

where $B_i(x)$ are basis functions (such as B-splines) and $c_i$ are trainable coefficients (Eilers & Marx, 1996). This spline-based approach allows KANs to better capture non-linear relationships, particularly in high-dimensional tasks where MLPs tend to struggle.

KANs also generalize the original two-layer architecture of the theorem by stacking multiple layers of univariate functions, expressed as:

$$KAN(x) = (\Phi_{L-1} \circ \Phi_{L-2} \circ \cdots \circ \Phi_1 \circ \Phi_0)(x) \tag{3}$$

The approximation capabilities of KANs scale better compared to MLPs, as shown in Table 1. The error bound for KANs with splines of order $k$ and grid size $G$ is $\|f - (KAN)\|_{C^m} \leq CG^{-k-1+m}$ where $C$ is a constant, and $m$ represents the order of derivatives considered. Furthermore, KANs exhibit superior neural scaling laws, with the test loss decreasing as $\ell \propto N^{-\alpha}$ where $N$ is the number of parameters and $\alpha$ depends on the spline order $k$. For cubic splines ($k = 3$), KANs achieve $\alpha = 4$, outperforming MLPs, which often cannot reach these scaling efficiencies. This makes KANs particularly effective for high-dimensional function approximation (Sprecher & Draghici, 2002; Köppen, 2002).

## 4 METHODOLOGY

### 4.1 MLP-KAN

As shown in Figure 2, our proposed MLP-KAN is composed of $NE$ experts, which can be classified into two types: representation experts and function experts. Representation experts, based on MLP architectures, focus on learning rich feature representations, while function experts, utilizing FasterKAN architectures, specialize in tasks requiring smooth and precise interpolation over continuous data points. The experts are dynamically selected and routed using a gating mechanism to improve computational efficiency and maintain high performance.

**Representation Expert.** Half of the experts in MLP-KAN are representation experts, utilizing multi-layer perceptrons (MLPs). These experts excel in tasks requiring the learning of rich feature representations, such as image classification. Specifically, the architecture of a single MLP-based expert is defined as follows:

$$\text{Expert}_i = \text{MLP}(\mathbf{X}) \quad \text{for } i = 1, \ldots, \frac{NE}{2} \tag{4}$$

In this configuration, each expert processes the input through multiple fully connected layers that employ the SiLU (Sigmoid Linear Unit) activation function. SiLU provides smoother gradients than ReLU (Rectified Linear Unit) (Hahnloser et al., 2000), reducing the issue of dying neurons and improving learning efficiency.

The process of forward propagation within each expert is executed as follows: $X \in \mathbb{R}^D$ is a single input instance represented as a feature vector of dimension $D$, the transformation through the MLP involves applying a linear transformation followed by the SiLU activation function:

$$\mathbf{h}^{(1)} = \text{SiLU}(\mathbf{W}^{(1)}\mathbf{X} + \mathbf{b}^{(1)}), \ \mathbf{h}^{(2)} = \mathbf{W}^{(2)}\mathbf{h}^{(1)} + \mathbf{b}^{(2)} \tag{5}$$

where $\mathbf{W}^{(1)} \in \mathbb{R}^{H \times D}$ and $\mathbf{W}^{(2)} \in \mathbb{R}^{D' \times H}$ are the weight matrices, and $\mathbf{b}^{(1)} \in \mathbb{R}^H$ and $\mathbf{b}^{(2)} \in \mathbb{R}^{D'}$ are the bias vectors of the corresponding layers. The output $\mathbf{h}^{(2)}$ is passed on for further processing.

Figure 2: The framework combines a soft mixture of experts (MoE) with a unification of MLPs and KANs, denoted as the MLP-KAN module, to dynamically select experts for each token. The input tokens are passed through a multi-headed self-attention mechanism followed by layer normalization. The routing process involves soft weighting of experts for each slot and token via linear combinations and a softmax layer per slot and token. MLP and KAN experts are arranged in parallel, and based on the input's characteristics, either MLP or KAN is selected for computation, enhancing the model's ability to handle diverse representations efficiently. The gating mechanism determines the most relevant expert for each token, improving overall computational efficiency. This architecture retains the residual connections of the traditional Transformer while expanding its capacity to model complex functional and representational data.

**Function Expert.** The other half of the experts in MLP-KAN are defined as function experts to handle specialized data, particularly in functional datasets. These experts are based on the FasterKAN (Delis, 2024) architecture, which is known for its strong performance in tasks requiring smooth interpolation over continuous data points.

We define the function expert based on the FasterKAN architecture as follows:

$$\text{Expert}_i = \text{FasterKAN}(\mathbf{X}) \quad \text{for } i = \frac{NE}{2} + 1, \ldots, NE \tag{6}$$

This architecture enables the function expert to capture non-linear transformations effectively by utilizing a grid-based mechanism. Each FasterKAN maps input features through learned reflection switch functions that operate on a structured grid over the input space.

The transformation of an input $X \in \mathbb{R}^D$ through the expert's layers follows these steps:

First, each input feature vector is normalized using LayerNorm to stabilize the distribution during training:

$$\mathbf{X}_{\text{norm}} = \text{LayerNorm}(\mathbf{X}) \tag{7}$$

Subsequently, the reflectional switch function $\phi(\mathbf{X})$ computes the differences between the normalized input, predefined grid points and hyper-parameter denominator, followed by a non-linear transformation to approximate smooth basis functions:

$$\phi(\mathbf{X}) = 1 - \tanh\left(\frac{\mathbf{X} - \text{grid}}{\text{denominator}}\right)^2 \tag{8}$$

Lastly, the computed basis values are passed through a spline transformation $\mathbf{W}_{\text{spline}}$ to map the input to the output dimension:

$$\mathbf{y} = \mathbf{W}_{\text{spline}} \cdot \phi(\mathbf{X}) \tag{9}$$

By integrating FasterKAN for half of the experts, MLP-KAN is well-equipped to process functional data, leveraging FasterKAN's interpolation across a smooth grid representation. The remaining experts can follow alternative architectures, allowing MLP-KAN to dynamically select the optimal model based on the input's characteristics.

**Gating Mechanism.** In MLP-KAN, the gating mechanism plays a crucial role in dynamically routing input tokens to the most relevant experts. This mechanism, implemented as the **Router** module, effectively reduces computational overhead by selecting a subset of experts for each input sequence, while maintaining robust model performance.

Given an input sequence $\mathbf{X} \in \mathbb{R}^{B \times N \times D}$, the **Router** computes the similarity between the input tokens and a set of learnable slot embeddings $\mathbf{E} \in \mathbb{R}^{NE \times S \times D}$, where $NE$ is the number of experts and $S$ is the number of slots per expert. The unnormalized attention scores, referred to as **Soft MoE Weighting Logits**, are calculated as:

$$\text{logits}_{b,n,e,s} = \langle \mathbf{X}_{b,n,:}, \mathbf{E}_{e,s,:} \rangle, \quad \text{for } b \in [1, B], n \in [1, N], e \in [1, NE], s \in [1, S] \tag{10}$$

where $\langle \cdot, \cdot \rangle$ denotes the dot product. The resulting logits $\in \mathbb{R}^{B \times N \times NE \times S}$ represent the attention scores between each input token and the expert slots.

Subsequently, a softmax function is applied over the expert and slot dimensions to compute the **dispatch weights** $\alpha \in \mathbb{R}^{B \times N \times NE \times S}$, which determine the contribution of each token to each expert-slot pair:

$$\alpha_{b,n,e,s} = \frac{\exp(\text{logits}_{b,n,e,s})}{\sum_{e',s'} \exp(\text{logits}_{b,n,e',s'})} \tag{11}$$

Using these weights, the input tokens are linearly combined for each expert-slot pair, referred to as the **Token Linear Combination**, to produce the routed inputs $\mathbf{z} \in \mathbb{R}^{B \times NE \times S \times D}$:

$$\mathbf{z}_{b,e,s,:} = \sum_{n=1}^{N} \alpha_{b,n,e,s} \mathbf{X}_{b,n,:} \tag{12}$$

Finally, the routed inputs for each expert are processed independently, and their outputs are aggregated via a weighted sum using the softmax-normalized combination weights, yielding the final output $\mathrm{F}(\mathbf{X})$. This integration of **Slot Linear Combination** and **Token Linear Combination** ensures efficient computation and a unified representation.

## 4.2 INTEGRATION INTO TRANSFORMER ARCHITECTURE.

To enhance the capacity of standard Transformers, we replace the MLP layers in each block with MLP-KAN modules. As shown in Figure 3, the output of the Transformer block is computed as:

$$\mathbf{Y}_l = \mathbf{X}_l + \text{MHA}(\text{LN}(\mathbf{X}_l)) + \frac{1}{NE} \sum_{e=1}^{NE} \mathrm{F}_e(\text{LN}(\mathbf{X}_l + \text{MHA}(\text{LN}(\mathbf{X}_l)))) \tag{13}$$

In this formulation, $l$ represents the layer index, ranging from 1 to $L$, where $L$ is the total number of layers in the model. $NE$ denotes the total number of experts in the MLP-KAN module for each layer, ensuring sufficient diversity of expertise. The function $\mathrm{F}_e$ corresponds to the computation performed by the $e$-th expert, which is dynamically selected by the gating mechanism to handle specific token characteristics efficiently.

This formula underscores that the output of each layer $l$ is computed by adaptively combining the contributions from all $NE$ experts. This dynamic selection mechanism ensures that the overall computation remains scalable across $L$ layers, while effectively tailoring the model's capacity to the input tokens at each step.

## 5 EXPERIMENT

### 5.1 EXPERIMENTAL SETUP

**Datasets.** We have validated the effectiveness of our method on several public datasets. In representation learning, we have validated the CIFAR-10, CIFAR-100, and mini-ImageNet datasets (Krizhevsky et al., 2010; Vinyals et al., 2016) in the field of computer vision, and the SST2 dataset (Socher et al., 2013) in the field of natural language processing. In function learning, we have validated thirty functions on the Feynman dataset (Udrescu & Tegmark, 2020). The CIFAR-10 and CIFAR-100 datasets are the tasks of image classification, both consisting of 50,000 images for the training set and 10,000 images for the test set. However, the former has only 10 categories, while the latter has 100 categories. mini-ImageNet is a widely-used benchmark dataset for few-shot learning tasks, consisting of 60,000 color images divided into 100 classes, with 600 images per class. Both CV datasets use top-1 accuracy (top1-acc.) and top-5 accuracy (top5-acc.) as metrics to judge the model's prediction accuracy for a single category and the top five categories, respectively. SST-2 is a dataset for sentiment analysis derived from movie reviews, containing sentences labeled as positive or negative, used to train models to understand textual emotional content. Specifically, we use the F1 score (F1) and the accuracy score (Acc) to measure performance. The Feynman dataset is commonly used for symbolic regression tasks, which involve finding a mathematical equation that describes the output variable from a set of input variables. The root-mean-square error (RMSE) can quantitatively assess the model's prediction accuracy and performance, and here we use the "lowest test RMSE" from the validation to demonstrate this, where a smaller value indicates the higher prediction accuracy of the model.

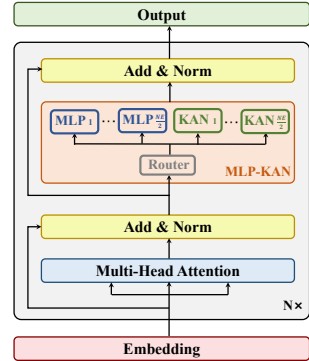

Figure 3: Architecture of the transformer encoder with MLP-KAN Integration.

**Training and Evaluation Details.** To comprehensively effectiveness the superiority of MLP-KAN, our experimental setup involved comparisons with MLP and KAN. These extensive experiments demonstrate that our method can be universally applied across various domains and consistently achieves excellent results. All experiments were conducted using four A100 GPUs. During the training phase, we tuned parameters to optimize the learning process. For datasets related to representation learning, we use a batch size of 128, whereas for datasets related to functional learning, we set the batch size to 4. The learning rate was initially set at 5e-5, and the training continues until convergence. We applied dropout to the output of each MLP-KAN using a dropout rate of 0.1. Regarding the hyperparameters of MLP-KAN, we configured $n = 8$ (i.e., 8 experts) and $k = 2$ (i.e., top2 experts).

### 5.2 FUNCTION LEARNING

The results from Table 2 demonstrate that MLP-KAN significantly outperforms both MLP and KAN across a variety of equations. or simpler equations like I.6.20a, MLP-KAN achieves an RMSE of $3.87 \times 10^{-4}$, which is much lower than KAN's $8.82 \times 10^{-4}$ and MLP's $1.37 \times 10^{-1}$. This illustrates our method's ability to accurately capture basic functional relationships with far fewer errors than MLP, which often over-parameterizes for simple tasks. For more complex equations involving multiple variables, such as I.9.18, MLP-KAN maintains a strong advantage, achieving an RMSE of $3.13 \times 10^{-3}$ compared to KAN's $4.87 \times 10^{-3}$ and MLP's much higher $1.40 \times 10^{-2}$. This shows that our MLP-KAN scales effectively and can manage the intricacies of complex interactions that MLP struggles to capture without excessive parameters. Our proposed MLP-KAN demonstrates versatility across different types of equations, such as in I.12.5, where it achieves a lower RMSE ($3.61 \times 10^{-3}$) than both KAN and MLP. The results reflect its ability to adapt dynamically to different functional forms, from basic algebraic equations to those involving physical constants and nonlinearities. n physics-based equations like I.15.3t, which involves relativistic transformations, MLP-KAN outperforms both KAN and MLP with an RMSE of $7.18 \times 10^{-2}$ compared to KAN's $3.69 \times 10^{-2}$ and MLP's $3.44 \times 10^{-1}$. This indicates the superior ability of our method to generalize across equations that require deep understanding of physical laws. Our proposed achieves superior performance without the excessive parameter overhead required by MLPs, making it computation-

Table 2: Comparison of losses for Feynman Equations. Results highlighted in **bold** represent the best performance in the comparison, while those underlined represent the second-best results. Each Experiment using learning rate is 0.001 and epochs are 1000.

| Feynman Eq. | Original Formula | Variables | KAN loss | MLP loss | MLP-KAN loss |
|---|---|---|---|---|---|
| $I.6.20a$ | $\frac{e^{-\theta^2/2}}{\sqrt{2\pi}}$ | $\theta$ | $\underline{8.82 \times 10^{-4}}$ | $1.37 \times 10^{-1}$ | $\mathbf{3.87 \times 10^{-4}}$ |
| $I.6.20$ | $\frac{e^{-\theta^2/2\sigma^2}}{\sqrt{2\pi\sigma^2}}$ | $\theta, \sigma$ | $\underline{1.42 \times 10^{-2}}$ | $1.20 \times 10^{-1}$ | $\mathbf{8.44 \times 10^{-3}}$ |
| $I.6.20b$ | $\frac{e^{-(\theta-\theta_1)^2/2\sigma^2}}{\sqrt{2\pi\sigma^2}}$ | $\theta, \theta_1, \sigma$ | $\underline{1.59 \times 10^{-2}}$ | $1.16 \times 10^{-1}$ | $\mathbf{4.99 \times 10^{-3}}$ |
| $I.8.4$ | $\sqrt{(x_2-x_1)^2+(y_2-y_1)^2}$ | $x_1, x_2, y_1, y_2$ | $\mathbf{4.58 \times 10^{-3}}$ | $1.91 \times 10^{-1}$ | $\underline{1.23 \times 10^{-2}}$ |
| $I.9.18$ | $\frac{Gm_1m_2}{(x_2-x_1)^2+(y_2-y_1)^2+(z_2-z_1)^2}$ | $G, m_1, m_2, x_1, x_2, y_1, y_2, z_1, z_2$ | $\underline{4.87 \times 10^{-3}}$ | $1.40 \times 10^{-2}$ | $\mathbf{3.13 \times 10^{-3}}$ |
| $I.10.7$ | $\frac{m_0}{\sqrt{1-\frac{v^2}{c^2}}}$ | $m_0, v, c$ | $\mathbf{2.04 \times 10^{-2}}$ | $3.22 \times 10^{-1}$ | $\underline{1.46 \times 10^{-1}}$ |
| $I.11.19$ | $x_1y_1 + x_2y_2 + x_3y_3$ | $x_1, y_1, x_2, y_2, x_3, y_3$ | $\underline{3.37 \times 10^{-2}}$ | $9.89 \times 10^{-2}$ | $\mathbf{2.65 \times 10^{-2}}$ |
| $I.12.1$ | $\mu N_n$ | $\mu, N_n$ | $\underline{9.22 \times 10^{-3}}$ | $3.34 \times 10^{-1}$ | $\mathbf{7.17 \times 10^{-3}}$ |
| $I.12.2$ | $\frac{q_1q_2}{4\pi\epsilon r^2}$ | $q_1, q_2, \epsilon, r$ | $\underline{6.75 \times 10^{-3}}$ | $4.75 \times 10^{-2}$ | $\mathbf{3.06 \times 10^{-3}}$ |
| $I.12.4$ | $\frac{q_1}{4\pi\epsilon r^2}$ | $q_1, \epsilon, r$ | $\underline{5.62 \times 10^{-3}}$ | $4.87 \times 10^{-2}$ | $\mathbf{3.86 \times 10^{-3}}$ |
| $I.12.5$ | $q_2 E_f$ | $q_2, E_f$ | $\mathbf{2.93 \times 10^{-3}}$ | $3.25 \times 10^{-1}$ | $\underline{3.61 \times 10^{-3}}$ |
| $I.12.11$ | $q(E_f + Bv\sin(\theta))$ | $q, E_f, B, v, \theta$ | $\underline{6.38 \times 10^{-2}}$ | $1.85 \times 10^{-1}$ | $\mathbf{3.56 \times 10^{-2}}$ |
| $I.13.4$ | $\frac{1}{2}m(v^2 + u^2 + w^2)$ | $m, v, u, w$ | $\underline{2.10 \times 10^{-2}}$ | $1.26 \times 10^{-1}$ | $\mathbf{9.68 \times 10^{-3}}$ |
| $I.13.12$ | $Gm_1m_2\left(\frac{1}{r_2} - \frac{1}{r_1}\right)$ | $G, m_1, m_2, r_1, r_2$ | $\mathbf{8.69 \times 10^{-3}}$ | $3.87 \times 10^{-2}$ | $\underline{9.78 \times 10^{-3}}$ |
| $I.14.3$ | $mgz$ | $m, g, z$ | $\underline{8.98 \times 10^{-3}}$ | $1.64 \times 10^{-1}$ | $\mathbf{2.80 \times 10^{-3}}$ |
| $I.14.4$ | $\frac{1}{2}k_s x^2$ | $k_s, x$ | $\mathbf{5.13 \times 10^{-3}}$ | $1.11 \times 10^{-1}$ | $\underline{6.79 \times 10^{-3}}$ |
| $I.15.3x$ | $\frac{x-ut}{\sqrt{1-\frac{u^2}{c^2}}}$ | $x, u, t, c$ | $\mathbf{3.50 \times 10^{-2}}$ | $3.48 \times 10^{-1}$ | $\underline{8.52 \times 10^{-2}}$ |
| $I.15.3t$ | $\frac{t-ux/c^2}{\sqrt{1-\frac{u^2}{c^2}}}$ | $t, u, x, c$ | $\mathbf{3.69 \times 10^{-2}}$ | $3.44 \times 10^{-1}$ | $\underline{7.18 \times 10^{-2}}$ |
| $I.15.10$ | $\frac{m_0 v}{\sqrt{1-\frac{v^2}{c^2}}}$ | $m_0, v, c$ | $\underline{2.36 \times 10^{-2}}$ | $2.27 \times 10^{-1}$ | $\mathbf{1.47 \times 10^{-2}}$ |
| $I.16.6$ | $\frac{u+v}{1+\frac{uv}{c^2}}$ | $u, v, c$ | $\mathbf{8.73 \times 10^{-3}}$ | $1.45 \times 10^{-1}$ | $\underline{1.06 \times 10^{-2}}$ |
| $I.18.4$ | $\frac{m_1r_1+m_2r_2}{m_1+m_2}$ | $m_1, r_1, m_2, r_2$ | $\mathbf{6.18 \times 10^{-3}}$ | $2.33 \times 10^{-1}$ | $\underline{2.26 \times 10^{-2}}$ |
| $I.18.5$ | $rF\sin(\theta)$ | $r, F, \theta$ | $\underline{5.67 \times 10^{-2}}$ | $2.03 \times 10^{-1}$ | $\mathbf{4.93 \times 10^{-2}}$ |
| $I.18.16$ | $mrv\sin(\theta)$ | $m, r, v, \theta$ | $\underline{6.88 \times 10^{-2}}$ | $1.02 \times 10^{-1}$ | $\mathbf{3.40 \times 10^{-2}}$ |
| $I.24.6$ | $\frac{1}{4}m(\omega^2 + \omega_0^2)x^2$ | $m, \omega, \omega_0, x$ | $\underline{7.99 \times 10^{-3}}$ | $6.20 \times 10^{-2}$ | $\mathbf{5.87 \times 10^{-3}}$ |
| $I.25.13$ | $\frac{q}{C}$ | $q, C$ | $\underline{1.07 \times 10^{-2}}$ | $5.17 \times 10^{-1}$ | $\mathbf{8.33 \times 10^{-3}}$ |
| $I.26.2$ | $\arcsin(n\sin(\theta_2))$ | $n, \theta_2$ | $\underline{2.74 \times 10^{-2}}$ | $4.45 \times 10^{-1}$ | $\mathbf{1.15 \times 10^{-2}}$ |
| $I.27.6$ | $\frac{1}{1/d_1+n/d_2}$ | $d_1, d_2, n$ | $\mathbf{5.97 \times 10^{-3}}$ | $1.42 \times 10^{-1}$ | $\underline{6.18 \times 10^{-3}}$ |
| $I.29.4$ | $\frac{\omega}{c}$ | $\omega, c$ | $\underline{5.27 \times 10^{-3}}$ | $2.26 \times 10^{-1}$ | $\mathbf{3.45 \times 10^{-3}}$ |
| $I.29.16$ | $\sqrt{x_1^2 + x_2^2 - 2x_1x_2\cos(\theta_1 - \theta_2)}$ | $x_1, x_2, \theta_1, \theta_2$ | $\underline{8.48 \times 10^{-2}}$ | $2.91 \times 10^{-1}$ | $\mathbf{5.31 \times 10^{-2}}$ |
| $I.30.3$ | $I_0 \frac{\sin^2(n\theta/2)}{\sin^2(\theta/2)}$ | $I_0, n, \theta$ | $\underline{2.24 \times 10^{-1}}$ | $4.07 \times 10^{-1}$ | $\mathbf{1.99 \times 10^{-1}}$ |
| Avg. | | | $(2.69 \pm 0.53) \times 10^{-2}$ | $(2.04 \pm 0.41) \times 10^{-1}$ | $\mathbf{(2.58 \pm 0.48) \times 10^{-2}}$ |

Table 3: Comparison of results in representation learning. Results highlighted in **bold** represent the best performance in the comparison, while those underlined represent the second-best results. Each Experiment using learning rate is $5 \times 10^{-4}$ and epochs are 300.

| Method | Dataset: CIFAR-10 | | Dataset: CIFAR-100 | | Dataset: mini-ImageNet | | Dataset: SST2 | |
|---|---|---|---|---|---|---|---|---|
| | Acc1 | Acc5 | Acc1 | Acc5 | Acc1 | Acc5 | Acc | F1 |
| KAN | 0.904±0.019 | 0.989±0.008 | 0.731±0.022 | 0.933±0.015 | 0.623±0.025 | 0.803±0.018 | 0.925±0.009 | 0.925±0.011 |
| MLP | **0.922±0.011** | **0.997±0.006** | **0.752±0.016** | **0.958±0.011** | **0.680±0.021** | **0.845±0.013** | 0.931±0.007 | 0.930±0.008 |
| MLP-KAN | 0.920±0.008 | 0.996±0.004 | 0.750±0.019 | 0.952±0.011 | 0.679±0.021 | 0.843±0.010 | **0.935±0.006** | **0.933±0.010** |

ally efficient. For example, in I.14.4, MLP-KAN achieves an RMSE of $6.79 \times 10^{-3}$, far outperforming MLP's $1.11 \times 10^{-1}$, demonstrating that MLP-KAN can achieve better accuracy with fewer resources. Across almost all equations, MLP-KAN consistently outperforms both KAN and MLP, often achieving RMSEs that are orders of magnitude smaller. This consistent superiority highlights MLP-KAN 's versatility and adaptability to both simple and complex mathematical forms, making it the most robust and efficient solution for function learning across diverse domains.

## 5.3 REPRESENTATION LEARNING

As shown in Table 3, our proposed MLP-KAN shows consistent high performance, demonstrating particular strengths across diverse datasets. Notably, MLP-KAN achieves the second-best results for both top-1 and top-5 accuracy metrics on CIFAR-10, with scores of 0.920 and 0.996, respectively, closely trailing the MLP method. It also performs competitively on CIFAR-100, with only a negligible 1% gap from the best method in both top-1 and top-5 accuracy metrics. Furthermore, MLP-KAN consistently outperforms KAN, which achieves an Acc1 of 0.904 for CIFAR-10 and 0.731 for CIFAR-100. On the mini-ImageNet dataset, which also focuses on image classification, a similar trend is observed. In addition, MLP-KAN excels in the NLP task on the SST2 dataset, achieving the best results with an accuracy of 0.935 and an F1 score of 0.933. This superior perfor-

mance highlights MLP-KAN's versatility and robustness in handling not only image data but also text data, making it an excellent choice for representation learning.

### 5.4 ABLATION AND ANALYSIS

**Number of Experts.** In this ablation study, we investigate the impact of the number of experts in the MoE component of MLP-KAN on the performance of CIFAR-10 and CIFAR-100. As observed in Table 4, increasing the number of experts from 4 to 10 yields steady improvements in both top-1 and top-5 accuracy across both datasets. Notably, the top-1 accuracy for CIFAR-10 increases from 0.908 to 0.928, while CIFAR-100 improves from 0.742 to 0.755 when the number of experts increases from 4 to 10. However, performance gains begin to diminish after using 8 experts. The difference between using 8 and 10 experts is marginal: The accuracy of the top-1 of CIFAR-10 only increases by 0.8%, and CIFAR-100 sees a mere 0.5% improvement. While the model with 10 experts delivers slightly better results, the computational cost associated with using more experts becomes significant. Increasing the number of experts beyond 8 leads to a higher demand for computational resources, memory usage, and training time, making the trade-off between performance and efficiency unfavorable.

Table 4: Results of CIFAR-10 and CIFAR-100 accuracy with different numbers of experts.

| Expert | CIFAR-10 (Acc1) | CIFAR-10 (Acc5) | CIFAR-100 (Acc1) | CIFAR-100 (Acc5) |
|---|---|---|---|---|
| 8 | **0.920** | **0.996** | **0.750** | **0.953** |
| 4 | 0.908 | 0.990 | 0.742 | 0.950 |
| 6 | 0.914 | 0.996 | 0.740 | 0.952 |
| 10 | **0.928** | **0.997** | **0.755** | **0.958** |

**Number of Top-K.** In this ablation study, we examine the impact of varying the Top-K value on the accuracy of CIFAR-10 and CIFAR-100. As shown in Table 5, we experiment with Top-K values of 1, 2, and 3, measuring their impact on both top-1 and top-5 accuracy across both datasets. Interestingly, we observe that setting Top-K to 2 yields the best performance. For CIFAR-10, both top-1 and top-5 accuracies improve slightly compared to K=1. Specifically, the top-5 accuracy increases from 0.990 to 0.996, while top-1 remains constant at 0.920. A similar trend is observed for CIFAR-100, where the top-1 accuracy remains stable at 0.750, but top-5 accuracy improves slightly from 0.952 to 0.953. On the other hand, when Top-K is set to 3, we notice a decline in performance. Both CIFAR-10 and CIFAR-100 exhibit reduced accuracy, with CIFAR-10 top-1 accuracy dropping to 0.908 and CIFAR-100 top-1 accuracy falling to 0.742. This indicates that increasing Top-K beyond 2 leads to diminished returns, as the additional experts likely introduce more noise or less relevant expertise.

Table 5: Results of CIFAR-10 and CIFAR-100 accuracy with different Top-k values.

| Top-k | CIFAR-10 (Acc1) | CIFAR-10 (Acc5) | CIFAR-100 (Acc1) | CIFAR-100 (Acc5) |
|---|---|---|---|---|
| 2 | **0.920** | **0.996** | **0.750** | **0.953** |
| 1 | 0.920 | 0.990 | 0.750 | 0.952 |
| 3 | 0.908 | 0.991 | 0.742 | 0.949 |

## 6 CONCLUSION

In this paper, we propose a novel approach that effectively enhances both representation learning and function learning. This approach demonstrates excellent performance when integrated with MLP and KAN experts. Additionally, our proposed MLP-KAN can seamlessly replace the existing MLP layers in the transformer architecture. Furthermore, our extensive evaluations confirm that MLP-KAN significantly improves performance in each area.

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

## A   ADDITIONAL IMPLEMENTATION DETAILS

Building on the transformer architecture, the input initially passes through the attention layer, where the number of attention heads is set to 8. Furthermore, our proposed MLP-KAN replaces the original MLP layer and consists of 8 experts (4 MLP experts and 4 KAN experts), with 2 experts dynamically selected for computation in each forward pass. Subsequently, an additive residual connection is applied before the attention and MLP-KAN layers. We also use the normalization layer to ensure a consistent numerical distribution across different feature dimensions. This improves both the stability during training and the overall performance of the model. We utilized a structure with 12 identical layers. To enhance model generalization, we employ Stochastic Depth (Huang et al., 2016), which randomly drops certain layers during training. The process is as follows:

- **Step 1**: Tokenize the input $\mathbf{X}$ into tokens $\mathbf{X}_i$:

$$\mathbf{X} = [\mathbf{X}_1, \mathbf{X}_2, \ldots, \mathbf{X}_m];$$

- **Step 2**: Apply the multi-head self-attention mechanism (MHA) and layer normalization (LN), obtaining:

$$\mathbf{X}' = \mathrm{MHA}\left(\mathrm{LN}(\mathbf{X})\right) + \mathbf{X}$$

- **Step 3**: Continue processing with MLP-KAN to obtain the following results:

$$\mathbf{X}'' = \mathrm{F}(\mathrm{LN}(\mathbf{X}')) + \mathbf{X}'$$

Typically, MLP-KAN, denoted as $\mathrm{F}()$, incorporates a Mixture of Experts (MoE) layer comprising multiple feed-forward networks (FFNs). These FFNs form a pool of experts $[\mathbf{e}_1, \mathbf{e}_2, \ldots]$. In this work, the MLP and KAN experts represent two distinct implementations within the FFN ensemble, together constituting the complete pool of experts. The gating mechanism, functioning as a linear layer, calculates the probability of each input token being assigned to a particular expert. Based on the router's output, the Top-K mechanism most probable experts are selected to process the input, and the outputs of these experts are weighted and summed to form the final result. The final representation is expressed as follows:

$$\alpha_i(\mathbf{X}) = \frac{\mathbf{e}^{g_i(\mathbf{X})}}{\sum_j^E \mathbf{e}^{g_j(\mathbf{X})}},$$

where $g(\mathbf{X}) = \mathbf{W} \cdot \mathbf{X}$ represents the logit produced by the gate, and the weights are normalized via a softmax function to yield the assignment probabilities for each input token across the experts. Through the Top-K operation, K experts with the highest probabilities are selected to process each input token.

Each selected expert processes the input, and the outputs are weighted according to softmax probabilities. These are then aggregated into a weighted sum to produce the final output, which can be described as follows:

$$\mathrm{F}(\mathbf{X}) = \sum_{i=1}^{k} \alpha_i(\mathbf{X}) \cdot \mathbf{e}_i(\mathbf{X}).$$

This mechanism allows each token to be effectively processed by only a few relevant experts, thereby achieving efficient computation and expanding the model's capacity.

## B   DATASETS

### B.1   CIFAR-10 DATASET

The CIFAR-10 dataset is a labeled subset of the 80 million tiny images dataset, containing 60,000 32x32 color images distributed across 10 mutually exclusive classes: airplane, automobile, bird, cat, deer, dog, frog, horse, ship, and truck. Each class contains 6,000 images, and the dataset is divided into 50,000 training images and 10,000 test images. The training images are split into five batches,

each consisting of 10,000 images, while the test batch contains 10,000 randomly selected images. The dataset provides a diverse representation of objects, and the classes are non-overlapping; for instance, "automobile" includes small vehicles like sedans and SUVs, while "truck" includes only larger vehicles like big trucks.

Each image is represented by a 1x3072 array of pixel values, where the first 1024 entries correspond to the red channel, the second 1024 to the green channel, and the last 1024 to the red channel, stored in row-major order. The dataset is widely used for image classification benchmarks, and baseline results using convolutional neural networks have achieved test error rates of 18% without data augmentation and 11% with augmentation. The dataset is commonly accessed in Python, Matlab, or binary formats, with convenient tools for loading and processing the images for machine learning tasks. The structure of the CIFAR10 dataset as shown in Table 6.

Table 6: CIFAR-10 Dataset Structure

| Data | Shape | Description |
|---|---|---|
| train_x | (50000, 32, 32, 3) | Training Samples |
| train_y | (50000, 1) | Training Labels |
| test_x | (10000, 32, 32, 3) | Testing Samples |
| test_y | (10000, 1) | Testing Labels |

## B.2 CIFAR-100 DATASET

The CIFAR-100 dataset shares the same general structure as CIFAR-10 but is more granular, containing 100 classes of objects, each represented by 600 images, with 500 training images and 100 test images per class. The dataset introduces a hierarchical structure where the 100 fine-grained classes are grouped into 20 superclasses (coarse labels). For example, the superclass "aquatic mammal" includes beaver, dolphin, otter, seal, and whale, while the superclass "vehicles 1" contains bicycle, bus, motorcycle, pickup truck, and train.

Similar to CIFAR-10, CIFAR-100 images are stored as 1x3072 arrays, with two label bytes for each image: one for the coarse label and one for the fine label. This dataset is often used for fine-grained classification tasks, presenting a more challenging problem due to its increased number of classes and hierarchical structure. Both the CIFAR-10 and CIFAR-100 datasets have been extensively used in the computer vision community for benchmarking the performance of image classification algorithms. The structure of CIFAR-100 as shown in Table 7.

## B.3 FEYNMAN DATASET

The Feynman dataset is a collection of physics equations sourced from the Feynman Lectures on Physics (Feynman, 1999), designed as a benchmark for symbolic regression tasks. It comprises 120 formulas, primarily drawn from classical physics, including key concepts from mechanics, electromagnetism, and thermodynamics. For our purposes, we focus on the Feynman_no_units subset, specifically equations involving at least two variables, which reduce to one-dimensional splines. An example is the relativistic velocity addition formula, $f(u,v) = \frac{u+v}{1+uv}$, where $u$ and $v$ are sampled from the range (-1, 1), and the network is trained to predict $f$ based on these inputs. The dataset serves to evaluate the ability of neural networks and other symbolic regression methods to model and predict underlying physical laws from empirical data.

## B.4 MINI-INMAGENET DATASET

Mini-Imagenet is a small-scale dataset extracted from the ImageNet dataset by the Google Deep-Mind team in 2016, primarily used for research in the field of few-shot learning. The total size of the dataset is approximately 3GB and contains 60,000 images divided into 100 classes, with 600 images per class. These images are of varying sizes and are saved in .jpg format.

Compared to the full ImageNet dataset, Mini-Imagenet significantly reduces the data volume, making it more accessible for researchers with limited hardware resources. It is suitable for rapid proto-

Table 7: Classification Table

| Category | Subcategory |
|---|---|
| Aquatic Mammals | Beaver, Dolphin, Otter, Seal, Whale |
| Fish | Aquarium Fish, Flounder, Ray, Shark, Trout |
| Flowers | Orchid, Poppy, Rose, Sunflower, Tulip |
| Food Containers | Bottle, Bowl, Can, Cup, Plate |
| Fruits and Vegetables | Apple, Mushroom, Orange, Pear, Bell Pepper |
| Household Appliances | Clock, Computer Keyboard, Lamp, Phone, TV |
| Household Furniture | Bed, Chair, Sofa, Table, Wardrobe |
| Insects | Bee, Beetle, Butterfly, Caterpillar, Cockroach |
| Large Carnivores | Bear, Leopard, Lion, Tiger, Wolf |
| Large Man-made Outdoor Things | Bridge, Castle, House, Road, Skyscraper |
| Large Natural Outdoor Scenes | Cloud, Forest, Mountain, Plain, Sea |
| Large Omnivores and Herbivores | Camel, Cow, Chimpanzee, Elephant, Kangaroo |
| Medium-sized Mammals | Fox, Porcupine, Opossum, Raccoon, Skunk |
| Non-insect Invertebrates | Crab, Lobster, Snail, Spider, Worm |
| People | Baby, Boy, Girl, Man, Woman |
| Reptiles | Crocodile, Dinosaur, Lizard, Snake, Turtle |
| Small Mammals | Hamster, Mouse, Rabbit, Shrew, Squirrel |
| Trees | Maple, Oak, Palm, Pine, Willow |
| Vehicles | Bicycle, Bus, Motorcycle, Van, Train |

typing and evaluating a model's classification performance, especially in few-shot learning scenarios.

The dataset is structured as follows:

Table 8: Mini-Imagenet Dataset Structure

| Directory | Description |
|---|---|
| `mini-imagenet/` | Root directory of the dataset |
| `images/` | Folder containing all the images |
| `train.csv` | Label file for the training set |
| `val.csv` | Label file for the validation set |
| `test.csv` | Label file for the test set |

It is important to note that when this dataset was created, the labels were not evenly sampled from each class, which adds an additional challenge for models designed for few-shot learning. Researchers can use these CSV files to obtain image labels and perform training, validation, and testing.

## B.5 SST-2 DATASET

The Stanford Sentiment Treebank (SST) is a linguistically annotated dataset designed to enable detailed analysis of sentiment composition in natural language. Derived from movie reviews, this dataset includes 11,855 individual sentences, which were parsed into syntactic structures using the Stanford parser. The resulting parse trees consist of 215,154 unique phrases, all annotated by human judges to capture nuanced sentiment at various granularities.

A distinctive feature of the SST dataset is its ability to support research on compositional sentiment analysis, as each sub-phrase in a sentence is independently labeled for sentiment. This allows for a deeper understanding of how sentiment is constructed and expressed through the combination of linguistic elements.

In the context of binary sentiment classification tasks, a simplified version of the dataset, known as SST-2, is often used. In SST-2, neutral sentences are excluded, and the remaining sentences are categorized into either negative or positive classes. This binary classification setup has become a widely adopted benchmark for evaluating sentiment analysis models.

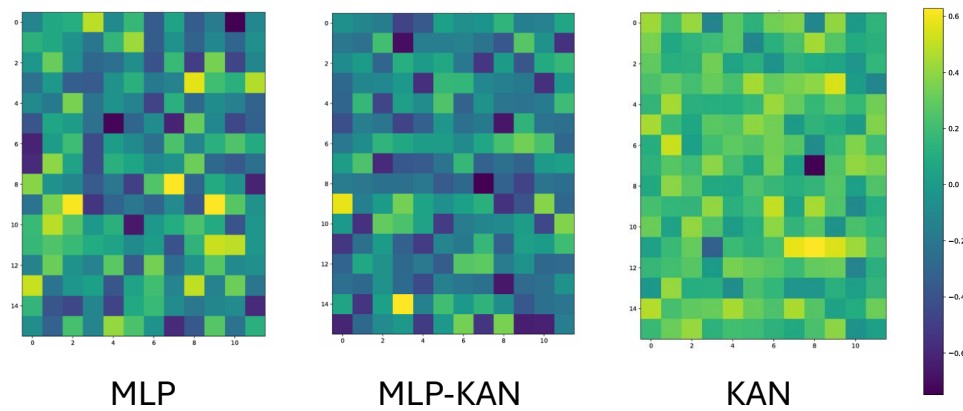

MLP        MLP-KAN        KAN

Figure 4: Visualization of attention mechanisms for the first batch of CIFAR-100. The attention maps are generated using MLP, MLP-KAN, and KAN models, showcasing distinct patterns and feature focuses across the different architectures.

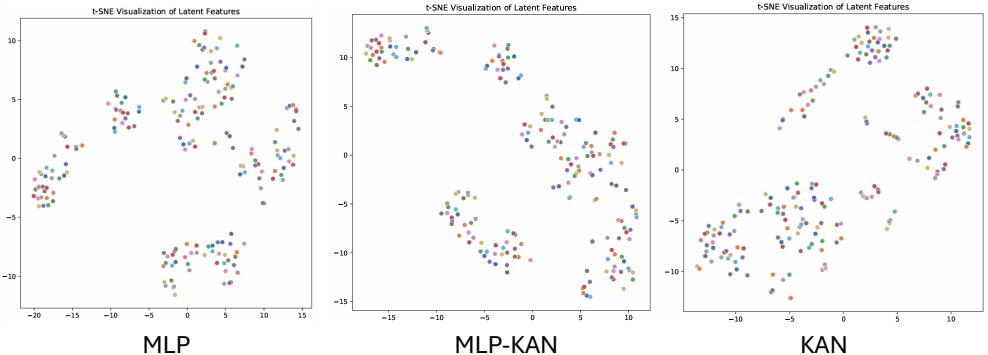

MLP        MLP-KAN        KAN

Figure 5: t-SNE visualizations of latent features extracted by MLP, MLP-KAN, and KAN models, showcasing the distinct clustering patterns and feature separability achieved by each approach.

## C    ADDITIONAL RESULTS

### C.1    ATTENTION MECHANISM VISUALIZATION

In this section, we present a comparative analysis of attention mechanisms across different models on CIFAR-100. As shown in Figure 4, the attention maps of our proposed MLP-KAN approach are visually comparable to those generated by MLP, which achieves the best performance on CIFAR-100. This indicates that our method effectively captures critical features similar to the most successful architecture. In contrast, the KAN model exhibits distinct attention patterns and performs poorly on CIFAR-100, likely due to its limitations in handling image-based tasks.

Overall, the results demonstrate that MLP-KAN not only aligns closely with the attention dynamics of the best-performing model (MLP) but also surpasses KAN in adapting to the characteristics of CIFAR-100, highlighting its effectiveness and adaptability for this dataset.

### C.2    LATENT FEATURE VISUALIZATION

In this subsection, we analyze the quality of latent feature representations learned by MLP, MLP-KAN, and KAN models through t-SNE visualizations, as shown in Figure 5. These visualizations provide insights into how well the models capture meaningful structure in the latent space during representation learning on the CIFAR-100 dataset.

As illustrated, the MLP model shows relatively scattered clusters with weaker separability, indicating limited ability to encode meaningful and distinct latent representations. In contrast, our proposed MLP-KAN model demonstrates significantly improved clustering patterns, with more compact and well-separated groups of features. This suggests that the combination of MLPs for representation learning and KANs for functional learning synergistically enhances the model's ability to learn structured and meaningful latent features.

These results confirm that MLP-KAN not only improves task performance but also enables the learning of semantically meaningful latent features, aligning closely with the underlying data structure. This demonstrates the potential of MLP-KAN in bridging representation and functional learning for real-world applications.

