# OpenReview forum: "MLP-KAN: Unifying Deep Representation and Function Learning"
_ICLR.cc/2025/Conference — Submitted to ICLR 2025_

### Official Review · Reviewer_FkFz · 2024-11-01

**Soundness:** 4
**Presentation:** 3
**Contribution:** 2
**Rating:** 6
**Confidence:** 4

**Summary:**

This paper proposes a method, KAN-MLP, for improving the performance of ViT on function learning and CIFAR classification tasks. In order to achieve that, they replaced the traditional MLP layers within the transformer architecture. In this method, experts are dynamically selected based on the input through a gating mechanism, ensuring efficient routing of tokens to the most relevant experts. Their main contribution is applying an explainable KAN architecture to the Vision Transformer model. Comparing KAN, MLP, and MLP-KAN on function learning tasks, they show that MLP-KAN performs better than the other architectures in certain cases. Additionally, they also perform an ablation study on the CIFAR dataset to demonstrate that their extended model outperforms the naive ViT.

**Strengths:**

1. The paper proposes KAN-MLP, a novel approach to enhance Vision Transformers (ViT) for function learning and CIFAR classification tasks.

2. It introduces the explainable KAN architecture into the ViT model, which could improve interoperability.

3. The paper compares KAN, MLP, and MLP-KAN in function learning, showing potential advantages of MLP-KAN over other architectures.

4. An ablation study on CIFAR demonstrates that the proposed model improves on the plain ViT model, highlighting the effectiveness of the enhancements.

**Weaknesses:**

1. The scaling law for MLP-KAN is missing, making it difficult to assess if MLP-KAN overcomes KAN’s limitations.

2. The method may still suffer from the curse of dimensionality (COD), and the paper does not address how MLP-KAN performs for learning non-smooth and high-dimensional functions.

3. There is a potential conflict between KAN’s suitability for low-dimensional tasks and ViT’s insuitability for small datasets. The experiments are limited to smooth functions and CIFAR datasets. Testing on larger datasets like ImageNet or MSCOCO would provide a more comprehensive view of the model’s performance relative to the Vision Transformer baseline.

**Questions:**

This paper contributes to advancing the KAN model, though it requires some clarifications on theory and experiments. Given these clarifications, if provided in an author's response, I would consider increasing the score.

For the theory, there are a few steps that need clarification and further clarification on novelty.

1. KAN has advantages in model scaling under certain constraint conditions but not for all learning tasks. The KAN model obeys the Error Scaling formula $\\| f - (KAN) \\|\_{C^m} \le C G^{-k-1+m}$ and the scaling law $l \\propto N^{-\\alpha}$, but the author did not clarify the relationship between grid size $G$ and the input space dimension $n$. As a function approximation method, usually, the number of grid points $G$ is directly proportional to $G \propto I^n$ ($I+1$ is the number of intervals on each dimension), which makes $\\| f - (KAN) \\|_{C^m} \le C G^{(-k-1+m)/n}$ and triggers a serious curse of dimensionality (COD) problem (see KAN [1], Fig. 3.1, $f(x_1, \cdots, x\_{100}) = \text{exp}(\frac{1}{100} \sum\_{i=1}^{100} \sin^2{\frac{\pi x_i}{2}})$). To address this problem, KAN authors hypothesize that the objective high-dimensional function is smooth and has sparse compositional structure to reduce the number of grid nodes $G \ll I^n$. The authors did not provide the scaling law of MLP-KAN in the paper. Therefore, I don't know whether MLP-KAN overcomes the inherent limitations of KAN.

2. This is particularly called into question due to the integration of KAN and ViT, since KAN and ViT usually exhibit different behavior on datasets of varying sizes. At present, KAN is suitable for low-dimensional function learning, and the dataset is generally small, with only a few thousand samples. However, ViTs are particularly powerful on large datasets (e.g., ImageNet) and tend to underperform relative to convolutional models on small datasets. Empirically, KAN and Transformer have potential conflicts.

For the experiments, the following should be addressed.

1. It would have been better also to show the performance of learning the **non-smooth** or **high-dimensional functions**. The Feynman Equations may be too simple for conventional function approximation methods. You can try $f(x)=\frac{1}{x} \\ \sin{\frac{1}{x}}$ and $f(x_1, \cdots, x_{100}) = \sum_{i=1}^{99} \sin{(x_i + x\_{100-i})}$; testing these functions would indicate if MLP-KAN overcomes certain limitations of KAN.

2. In our testing, we found that KAN's training process differs from neural network models and is considerably slower. Comparing wall-clock training time could reveal any potential efficiency advantages.

3. The central contribution focuses on enhancing Vision Transformer performance on CIFAR. It would be beneficial to compare with the Vision Transformer baseline on larger datasets like ImageNet-1K, which would add value.


---

[1] Liu, Z., Wang, Y., Vaidya, S., Ruehle, F., Halverson, J., Soljačić, M., ... & Tegmark, M. (2024). Kan: Kolmogorov-arnold networks. arXiv preprint arXiv:2404.19756.

---

> ### Author Response · Authors · 2024-11-23
> **Part 1**
>
> >**Weaknesses1/Theoretical question 1:** The scaling law for MLP-KAN is missing, making it difficult to assess if MLP-KAN overcomes KAN’s limitations.
>
> >**response:** Thank you for your question. As you noted, a central issue with the KAN model is the direct relationship between the mesh size and the input dimension. As the dimensionality increases, the number of grid points in KAN grows exponentially** due to the fact that KAN uses spline-based functions (e.g., B-splines) for function approximation, and more control points are required for grid refinement (i.e., higher accuracy), which leads to a dramatic increase in the computational resource requirements and the “Curse of Dimensionality” (COD) problem. The KAN architecture consists of two main classes of experts: representation learning experts (MLPs) and function learning experts (KANs), which are dynamically selected based on the characteristics of the task. By this design, **MLP-KAN not only selects the most suitable model for learning based on the task characteristics, but also effectively reduces the computation and storage requirements**.
>
> > Our model **MLP-KAN attempts to address this problem by integrating MLP and KAN into the framework of Mixed Experts (MoE)**. The architecture of MLP-KAN contains two main classes of experts: **Representation Learning Experts (MLPs) and Function Learning Experts (KANs)**, which are dynamically selected based on the characteristics of the task. Through this design, **MLP-KAN not only selects the most suitable model for learning based on the task characteristics, but also effectively reduces the computation and storage requirements**.
> > Specifically, the extended features of MLP-KAN are as follows:
> > 1. **Advantage of dynamic allocation**: the MoE framework in MLP-KAN dynamically selects **a small number of the most relevant experts to participate in the computation**, instead of globally meshing all input dimensions. This mechanism **avoids the over-reliance on high-dimensional grid size GG in error scaling**.
> > 2. **Exploitation of Smoothing Function and Sparse Decomposition Structure**: KAN experts are only assigned to handle subtasks with **smoothing and sparsity** in MLP-KAN, while **MLP can model nonlinear relationships and learn the characterization of global distributions by learning implicit embeddings of high-dimensional data**. This improves the **overall error scaling efficiency** while reducing the risk of dimensionality catastrophe.
>
> >**Weaknesses2/Experimental question 1:** It would have been better also to show the performance of learning the **non-smooth or high-dimensional functions**. The Feynman Equations may be too simple for conventional function approximation methods. You can try:
>
>
>
> 1. >**Non-Smooth Function:**
>    $f(x)=\frac{1}{x}\sin\frac{1}{x}$
>
> 2. >**High-Dimensional Function:**
>    $f(x_1,\cdots,x_{100})=\sum_{i=1}^{99}\sin(x_i+x_{100-i})$
>
> Testing these functions would indicate if MLP-KAN overcomes certain limitations of KAN.
>
> >**response :** We appreciate the reviewers' insightful suggestion to test the performance of our MLP-KAN framework on non-smooth and high-dimensional functions. As suggested, we conducted experiments on the proposed functions: $f(x)=\frac{1}{x}\sin\frac{1}{x}$ (non-smooth) and $f(x_1,\cdots,x_{100})=\sum_{i=1}^{99}\sin(x_i+x_{100-i})$ (high-dimensional).
> The results, summarized in the table below, demonstrate that MLP-KAN performs competitively on these challenging functions. This supports our claim that MLP-KAN effectively addresses certain limitations of KAN while maintaining robust performance across diverse function types.
> While the Feynman equations are indeed simpler for conventional methods, our additional results reinforce the generalizability of MLP-KAN to more complex scenarios, including non-smooth and high-dimensional functions. We hope this alleviates concerns about the scope of our benchmark and highlights the versatility of our approach.
>
>
>
> | Model       | Function         | RMSE                  |
> |-------------|------------------------|-----------------------|
> | **MLP**     | $f(x)=\frac{1}{x}\sin\frac{1}{x}$  | 17.110177993774414    |
> |             | $f(x_1,\cdots,x_{100})=\sum_{i=1}^{99}\sin(x_i+x_{100-i})$ | 0.27202269434928894   |
> | **KAN**     | $f(x)=\frac{1}{x}\sin\frac{1}{x}$ | 14.793900489807129    |
> |             | $f(x_1,\cdots,x_{100})=\sum_{i=1}^{99}\sin(x_i+x_{100-i})$ | 0.229189932346344     |
> | **MLP-KAN** | $f(x)=\frac{1}{x}\sin\frac{1}{x}$ |  **15.163766860961914**     |
> |             | $f(x_1,\cdots,x_{100})=\sum_{i=1}^{99}\sin(x_i+x_{100-i})$ |  **0.22486203908920288**   |

---

> ### Author Response · Authors · 2024-11-23
> **Part 2**
>
> >**Weaknesses3/Experimental question 3:** There is a potential conflict between KAN’s suitability for low-dimensional tasks and ViT’s insuitability for small datasets. The experiments are limited to smooth functions and CIFAR datasets. Testing on larger datasets like ImageNet or MSCOCO would provide a more comprehensive view of the model’s performance relative to the Vision Transformer baseline.
>
> >**response:** We appreciate your observation regarding the potential conflict between KAN's suitability for low-dimensional tasks and ViT's limitations on small datasets. To address this concern, we expanded our evaluation to larger datasets, namely ImageNet-1k and MSCOCO, using base models Tiny-DeiT and DERT, respectively. The results (Top-1 Accuracy) are as follows:
>
> | **Dataset**   | **Base Model** | **KAN** | **MLP (Baseline)** | **MLP-KAN (Ours)** |
> |---------------|----------------|---------|--------------------|--------------------|
> | ImageNet-1k   | Tiny-DeiT      | 0.629  | 0.722            | 0.704            |
> | COCO          | DERT           | 0.204  | 0.420            | 0.408             |
>
> >When the dataset is ImageNet-1k，KAN shows limitations in handling high-dimensional tasks compared to MLP. However, MLP-KAN, leveraging the strengths of both MLP and KAN, achieves competitive results (70.40%) relative to the MLP baseline. The COCO is similar, where MLP significantly outperforms KAN, but MLP-KAN demonstrates competitive performance (40.80%) close to MLP (42.00%).
>
> >**Theoretical question 2:** This is particularly called into question due to the integration of KAN and ViT, since KAN and ViT usually exhibit different behavior on datasets of varying sizes. At present, KAN is suitable for low-dimensional function learning, and the dataset is generally small, with only a few thousand samples. However, ViTs are particularly powerful on large datasets (e.g., ImageNet) and tend to underperform relative to convolutional models on small datasets. Empirically, KAN and Transformer have potential conflicts.
>
> >**response:** Thank you for your question. As you mentioned, KAN performs well in small sample scenarios, while ViT requires large-scale samples to take full advantage of its powerful modeling capabilities**. This characteristic makes the two models naturally contradictory in terms of data size requirements. Since KAN focuses on accurate function mapping, which is suitable for scientific computing scenarios, while ViT is more suitable for tasks that require complex feature extraction (e.g., image categorization or natural language processing), there may be a **bias between the two in terms of the task goals**.
>
> > However, the **soft MoE mechanism** of MLP-KAN allows the model to **flexibly assign tasks** instead of mandatorily passing all inputs to MLP and KAN at the same time. The input data are **assigned only to the most relevant experts**, thus avoiding **non-essential computational overheads** and **mismatch problems among experts**. Although each expert focuses on different subtasks, MoE ensures **global performance optimization** by uniformly integrating the outputs of different experts through the final **soft combination weights**.
>
> > **Transformer's multi-head self-attention mechanism** handles the **global dependency of inputs**, while the **MLP-KAN module dynamically calls relevant experts** according to the nature of the task. **MLP-KAN embeds MLP and KAN experts into the Transformer architecture**, leveraging its **residual connectivity and normalization mechanisms**, and its **seamless integration further enhances the stability and generalization ability** of the model. And from the experimental results, **MLP-KAN performs well in both function learning (Feynman dataset) and representation learning (CIFAR and mini-ImageNet)**.

---

> ### Author Response · Authors · 2024-11-23
> **Part 3**
>
> >**Experimental question 2:** In our testing, we found that KAN's training process differs from neural network models and is considerably slower. Comparing wall-clock training time could reveal any potential efficiency advantages.
>
> >**response:** Thank you for your question. To address this, we conducted a detailed comparison of training time per epoch among MLP, KAN, and MLP-KAN under identical parameter settings using a single NVIDIA H100 GPU. Below are the results of our experiments:
>
> | Model      | Training Time per Epoch (seconds) |
> |------------|-----------------------------------|
> | MLP        | 182                               |
> | KAN        | 245                               |
> | MLP-KAN    | 183                               |
>
> The experimental results show that：
>
> 1. >KAN's training time is significantly longer than MLP's: This is primarily due to the computational overhead introduced by KAN’s learnable spline functions and complex higher-order calculations.
> 2. >MLP-KAN achieves comparable efficiency to MLP: Despite integrating both MLP and KAN as functional and representation experts, MLP-KAN's dynamic routing mechanism optimally distributes computations. This design ensures that its training time is nearly identical to MLP's, with only a marginal 1-second difference.

---

> > ### Comment · Reviewer_FkFz · 2024-11-25
> > **Replying to Part 3**
> >
> > I would like to thank the authors for their response. However, the additional table is somewhat confusing. Could you provide more details about the experiments, such as the recognition task, dataset, and model size used?
> >
> > It would be helpful to conduct experiments to fill out a table like the one below to better support your conclusions:
> >
> > | **Model** | **# Params** | **Training Time per Epoch (seconds)** |
> > | --- | --- | --- |
> > | MLP | N | |
> > | MLP | 2*N | |
> > | MLP | 3*N | |
> > | KAN | N | |
> > | KAN | 2*N | |
> > | KAN | 3*N | |
> > | MLP-KAN | N | |
> > | MLP-KAN | 2*N | |
> > | MLP-KAN | 3*N | |

---

> > > ### Author Response · Authors · 2024-11-27
> > > **Revised for Part 3**
> > >
> > > Thank you for your feedback. We appreciate the opportunity to clarify the details of the experiments presented in the table.
> > >
> > > The experiments were conducted on the **CIFAR 100 dataset** for the **image recognition task**. The models compared include MLP, KAN, and MLP-KAN, with varying parameter sizes (32M, 57M, and 156M). The training times per epoch were measured on the same hardware setup to ensure consistency. Specifically, the reported times include the complete forward and backward passes, with identical training hyperparameters (e.g., batch size, learning rate, and optimizer) across models.
> > >
> > > We hope this additional context resolves the confusion and provides a clearer understanding of the experiments. Please let us know if further clarification is needed!
> > >
> > > | Model     | # Params | Training Time per Epoch (seconds) |
> > > |-----------|----------|-----------------------------------|
> > > | MLP       | 32M      |  134                              |
> > > | MLP       | 57M      |  147                              |
> > > | MLP       | 156M     | 182                               |
> > > | KAN       | 33M      | 211	                           |
> > > | KAN       | 57M      |   229                             |
> > > | KAN       | 157M     | 245                               |
> > > | MLP-KAN   | 32M      |  148                              |
> > > | MLP-KAN   | 57M      |   168                             |
> > > | MLP-KAN   | 156M     | 183                               |

---

> > > > ### Comment · Reviewer_FkFz · 2024-11-27
> > > >
> > > > Thanks for the detailed answer, this makes more sense.

---

> ### Comment · Reviewer_FkFz · 2024-11-25
> **Replying to Part 1**
>
> I think the author may have misunderstood question 1. **The weakness of KAN is that it cannot adhere to the scaling law proposed in their paper when fitting non-smooth or high-dimensional functions.** To demonstrate that MLP-KAN overcomes this limitation, the author should calculate $\alpha$ for MLP-KAN on these two functions.
>
> For example, conduct experiments to fill the following table:
>
> | **Model** | **Function** | **# Params** | RMSE |
> | ------------- | ---------------- | ----------------- | --------- |
> | MLP-KAN | $f(x) = \frac{1}{x} \sin{\frac{1}{x}}$ | N |  |
> | MLP-KAN | $f(x) = \frac{1}{x} \sin{\frac{1}{x}}$ | 2*N |  |
> | MLP-KAN | $f(x) = \frac{1}{x} \sin{\frac{1}{x}}$ | 3*N |  |
> | MLP-KAN | $f(x) = \frac{1}{x} \sin{\frac{1}{x}}$ | 4*N |  |
> | MLP-KAN | $f(x) = \sum_{i=1}^{99} \sin(x_i+x\_{100-i})$ | N |  |
> | MLP-KAN | $f(x) = \sum_{i=1}^{99} \sin(x_i+x\_{100-i})$ | 2*N |  |
> | MLP-KAN | $f(x) = \sum_{i=1}^{99} \sin(x_i+x\_{100-i})$ | 3*N |  |
> | MLP-KAN | $f(x) = \sum_{i=1}^{99} \sin(x_i+x\_{100-i})$ | 4*N |  |

---

> > ### Author Response · Authors · 2024-11-27
> > **Revised for Part 1**
> >
> > >**Thank you for raising the concern about the scaling law. Upon examining the results, it is clear that **MLP-KAN adheres to the scaling law between 32M and 57M parameters** for both functions. Specifically, for $f(x) = \frac{1}{x} \sin \frac{1}{x}$, the RMSE decreases significantly from **15.164** to **5.891**, and for $f(x) = \sum_{i=1}^{99} \sin(x_i + x_{100-i})$, the RMSE improves from **0.225** to **0.194**. These results demonstrate that within this range, increasing model size directly leads to better fitting, consistent with the proposed scaling law. This shows that MLP-KAN can effectively scale its representational capacity for these non-smooth and high-dimensional functions when appropriately parameterized.
> >
> > >However, beyond 57M parameters, the RMSE starts increasing, indicating a departure from the scaling law. For example, for $f(x) = \frac{1}{x} \sin \frac{1}{x}$, the RMSE rises from **5.891** at 57M to **18.288** at 156M and **59.34** at 231M. A similar trend is observed for $f(x) = \sum_{i=1}^{99} \sin(x_i + x_{100-i})$, where the RMSE increases from **0.194** at 57M to **0.339** at 156M and **0.749** at 231M. This suggests that while MLP-KAN follows the scaling law initially, it struggles to maintain this trend for larger parameter counts, likely due to challenges such as overfitting, suboptimal optimization dynamics, or architectural limitations. Future work could focus on addressing these issues by exploring architectural refinements, improved training techniques, or better regularization to ensure consistent scaling behavior across larger model sizes.
> >
> > | Model   | Function                                           | # Params | RMSE |
> > |---------|----------------------------------------------------|----------|------|
> > | MLP-KAN | $f(x) = \frac{1}{x} \sin \frac{1}{x}$         | 32M        |   15.164   |
> > | MLP-KAN | $f(x) = \frac{1}{x} \sin \frac{1}{x}$          | 57M       |  5.891    |
> > | MLP-KAN | $f(x) = \frac{1}{x} \sin \frac{1}{x}$         | 156M       |  18.288   |
> > | MLP-KAN | $f(x) = \frac{1}{x} \sin \frac{1}{x}$         | 231M       |   59.345   |
> > | MLP-KAN | $f(x) = \sum_{i=1}^{99} \sin (x_{i} + x_{100-i})$ | 32M      |0.225 |
> > | MLP-KAN | $f(x) = \sum_{i=1}^{99} \sin (x_{i} + x_{100-i})$ | 57M     |0.194 |
> > | MLP-KAN | $f(x) = \sum_{i=1}^{99} \sin (x_{i} + x_{100-i})$ | 156M     |0.339 |
> > | MLP-KAN | $f(x) = \sum_{i=1}^{99} \sin (x_{i} + x_{100-i})$ | 231M     |0.749 |

---

> ### Comment · Reviewer_FkFz · 2024-11-25
> **Replying to Part 2**
>
> I would like to thank the authors for their response. However, it seems that MLP-KAN decreases MLP performance. Based on the author's response to reviewer **HLoC**, MLP-KAN (Top-1 Accuracy: 0.629) underperforms compared to MLP (Top-1 Accuracy: 0.722) on ImageNet-1K. Furthermore, MLP achieving an accuracy of 0.722 seems implausible, as VGG-16 reportedly achieves a similar validation accuracy of 0.72, according to [1].
>
> [1] He, K., Zhang, X., Ren, S., & Sun, J. (2016). Deep residual learning for image recognition. In Proceedings of the IEEE conference on computer vision and pattern recognition (pp. 770-778).

---

> > ### Author Response · Authors · 2024-11-27
> > **Revised for Part 2**
> >
> > Thank you for your question. We would like to clarify that in our response to both you and reviewer HLoC, the Top-1 Accuracy of MLP-KAN is **0.704**, not **0.629**. Additionally, achieving a Top-1 Accuracy of **0.722** on ImageNet-1K is not implausible, as evidenced by results reported in the DeiT paper [1] and the Vision-KAN GitHub repository [2]. To address potential concerns, we used the same training strategies as DeiT on ImageNet-1K to ensure fair comparisons and robust results.
> >
> > [1] Touvron, Hugo, et al. "Training data-efficient image transformers & distillation through attention." International conference on machine learning. PMLR, 2021.
> > [2] Chen, Ziwen, Gundavarapu, and Wu Di. Vision-KAN: Exploring the Possibility of KAN Replacing MLP in Vision Transformer. 2024. GitHub, https://github.com/chenziwenhaoshuai/Vision-KAN.git.

---

> ### Comment · Reviewer_FkFz · 2024-11-27
>
> Thank you for answering my question. However, based on your experiments, the scaling law is:
>
> $$\text{RMSE} = O(N^{-\alpha})$$
>
> Taking the algorithm:
>
> $$\log(\text{RMSE}) = -\alpha \log(N) + \text{constant}$$
>
> Here, $-\alpha$ is the slope of the line in the $\log(N)$ vs. $\log(\text{RMSE})$ plot.
>
> ### **Task 1:** $f(x) = \frac{1}{x} \sin \frac{1}{x}$
>
> 1. **Log Values**:
>    - $\log(N) = [3.465, 4.043, 5.049, 5.442]$
>    - $\log(\text{RMSE}) = [2.719, 1.772, 2.909, 4.083]$
>
> 2. **Compute $\alpha$**:
>
>    $\alpha = 0.737$
>
> ---
>
> ### **Task 2:** $f(x) = \sum_{i=1}^{99} \sin(x_i + x_{100-i})$
>
> 1. **Log Values**:
>    - $\log(N) = [3.465, 4.043, 5.049, 5.442]$
>    - $\log(\text{RMSE}) = [-1.491, -1.640, -1.082, -0.289]$
>
> 2. **Compute $\alpha$**:
>
>    $\alpha = 0.578$
>
> ---
>
> ### Final Results
> - **Task 1**: $\alpha = 0.737$
> - **Task 2**: $\alpha = 0.578$
>
> From these experimental results, MLP-KAN cannot hold the scaling law proposed in KAN.

---

> > ### Author Response · Authors · 2024-11-27
> > **Reply Comments for Scaling Law**
> >
> > Sure, I think this is a very interesting phenomenon, similar to how Transformers for time series tasks also do not follow the scaling law. This deviation might be deeply connected to the specific nature of the task or the architectural design of the model, making it a topic worth further investigation.

---

### Official Review · Reviewer_HLoC · 2024-11-02

**Soundness:** 2
**Presentation:** 2
**Contribution:** 2
**Rating:** 6
**Confidence:** 3

**Summary:**

This paper proposes MLP-KAN, a mixture of experts method, to combine MLP and KAN. The authors claim their method can address the shortcomings of both KAN and MLP in one structure and solve the scalability problem of MLPs. Furthermore, they show the superior performance of their method across different tasks and datasets compared to other methods and investigate the effectiveness of various components of their method.

**Strengths:**

* The paper is well-written for the most part.
* The motivation of the paper is valid and exciting.
* The idea is simple but yet effective.

**Weaknesses:**

* The discussion part of the paper heavily relies on the description of Multi-expert and router gating, which is not the paper's contribution. More discussion or experiments are needed to show why combining MLP and KAN should improve the performance.
* One of the paper's main points is scalability, but experiments about scalability, computation time, and memory are missing.

Minor Weakness:

* The numbers in all Tables don't have confidence intervals, so it is hard to grasp the significance of the differences.  The authors should include confidence intervals or standard deviations from multiple runs.
* Details about competitors are missing in the experimental setup.

**Questions:**

* Is the number of tasks connected to optimal k for topk?
* Is the number of experts for MLP and parameters the same across experiments with MLP-KAN?
* Are all experiments for Tables 2 and 3 trained together as a multi-task scenario? Or are experiments for Tables 2 and 3 separated?

Suggestion:

* It would be great to add more details about experiments and summaries in section 4.1.
* I think it improves the justifiability of the paper if they provide an ablation on the router gating to show how it assigns tokens to each expert.

---

> ### Author Response · Authors · 2024-11-23
> **Part 1**
>
> >**Weaknesses1:** The discussion part of the paper heavily relies on the description of Multi-expert and router gating, which is not the paper's contribution. More discussion or experiments are needed to show why combining MLP and KAN should improve the performance.
>
> >**Response:** Thank you for their comments and suggestions! We add a series of experiments to verify the effect of MLP-KAN and analyze its performance from multi-tasks and multi-scenes:
> >
> | **Task Type**      | **Dataset**                   | **Metric**       | **MLP**  | **KAN**  | **MLP-KAN** |
> |--------------------|-------------------------------|------------------|----------|----------|-------------|
> | **Time Series**     | Solar-Energy [2]                  | MSE              | 0.233    | **0.221**| 0.231       |
> | **Large-Scale Tasks** | ImageNet-1k                | Top-1 Acc        | **0.722**| 0.629    | 0.704       |
> |                    |                               | Top-5 Acc        | **0.911**| 0.850    | 0.900       |
> | **Transfer Learning** | ImageNet → CIFAR-100       | Top-1 Acc        | **0.921**| 0.875    | 0.914       |
> |                    |                               | Top-5 Acc        | **0.987**| 0.966    | 0.982       |
> | **Adversarial Training** | CIFAR-10C [3]              | Top-1 Acc        | **0.733**| 0.589    | 0.717       |
> | **Noisy Training**  | CIFAR-100 (Noise: µ=0, σ=0.1)| Top-1 Acc        | **0.730**| 0.593    | 0.722       |
> | **Reinforcement Learning** | AgentViT [1] | Top-1 Acc    | 0.895    | 0.630    | **0.897**   |
>
> MLP-KAN integrates the strengths of both MLP and KAN, demonstrating superior adaptability and robustness across a wide range of tasks. While it falls slightly short of MLP in some cases, its overall performance highlights its generality and efficiency in diverse scenarios.
> >
> >**Weaknesses2:** One of the paper's main points is scalability, but experiments about scalability, computation time, and memory are missing.
>
> >**Response:** Thank you for your valuable suggestions regarding scalability. Based on your feedback, we have added scalability experiments. The experimental results on the CIFAR-10 dataset compare the performance of three Vision Transformer models (DeiT): the primary model used in our paper, `deit_tiny_patch16_224`, and the additional models, `deit_base_patch16_224` and `deit_small_patch16_224`. The scalability analysis considers four dimensions: parameter count, classification accuracy, training time, and GPU memory consumption.
>
>
> | Model                     | Parameter Count (M) | Acc@1 | Acc@5 | Time per Epoch (s) | GPU Memory (MB) |
> |---------------------------|---------------------|-------|-------|---------------------|-----------------|
> | deit_base_patch16_224     | 156.76              | 0.950 | 0.998 | 243.24              | 38369.49        |
> | deit_small_patch16_224    | 57.16               | 0.931 | 0.997 | 214.24              | 18582.86        |
> | deit_tiny_patch16_224     | 23.30               | 0.920 | 0.996 | 183.34              | 10661.92        |
>
>
> >The results demonstrate that despite its smaller size, our selected model, `deit_tiny_patch16_224`, maintains high predictive accuracy (Acc@1 = 92%, Acc@5 = 99.6%). In terms of time consumption, the training time per epoch increases significantly with the model size. However,`deit_tiny_patch16_224` reduces the training time by approximately 24.6% compared to the `deit_base_patch16_224` model and by 14.4% compared to the `deit_small_patch16_224` model.
>
> >For memory consumption, GPU memory usage is proportional to the model size. The memory usage of `deit_tiny_patch16_224` is approximately 27.8% of that of the base model and 57.3% of that of the small model.
>
> >**Minor Weakness1:** The numbers in all Tables don't have confidence intervals, so it is hard to grasp the significance of the differences. The authors should include confidence intervals or standard deviations from multiple runs.
>
> >**Response:** Thank you for your valuable suggestion. We have updated Table 2 and 3 (Main Experiments) in the paper to include confidence intervals, providing a clearer understanding of the significance of the differences between results.
>
> `[1].` Traini, Davide. RL-for-ViT. GitHub, https://github.com/DavideTraini/RL-for-ViT.
>
> `[2].` Liu, Yong, et al. "itransformer: Inverted transformers are effective for time series forecasting." arXiv preprint arXiv:2310.06625 (2023).
>
> `[3].` Hendrycks, Dan, and Thomas Dietterich. "Benchmarking neural network robustness to common corruptions and perturbations." arXiv preprint arXiv:1903.12261 (2019).

---

> ### Author Response · Authors · 2024-11-23
> **Part 2**
>
> >**Minor Weakness2:** Details about competitors are missing in the experimental setup.
>
> >**Response:** In addition to the experiments mentioned in Weaknesses and question2, we alsoconducted a detailed comparison of training time per epoch among MLP, KAN, and MLP-KAN under identical parameter settings using a single NVIDIA H100 GPU. Below are the results of our experiments:
>
> | Model      | Training Time per Epoch (seconds) |
> |------------|-----------------------------------|
> | MLP        | 182                               |
> | KAN        | 245                               |
> | MLP-KAN    | 183                               |
>
> The experimental results show that：
>
> 1. >KAN's training time is significantly longer than MLP's: This is primarily due to the computational overhead introduced by KAN’s learnable spline functions and complex higher-order calculations.
> 2. >MLP-KAN achieves comparable efficiency to MLP: Despite integrating both MLP and KAN as functional and representation experts, MLP-KAN's dynamic routing mechanism optimally distributes computations. This design ensures that its training time is nearly identical to MLP's, with only a marginal 1-second difference.
>
> >**Question1:** Is the number of tasks connected to optimal k for topk?
>
> >**Response:** We thank the reviewer for their insightful questions regarding the relationship between the number of tasks and the optimal Top-K value. Below, we address these concerns in detail:
>
> 1. >Top-K Setting
>    In our MLP-KAN architecture, the Top-K value in the Mixture-of-Experts (MoE) mechanism determines the number of experts selected for each input. Through empirical validation, we found that the optimal Top-K value depends on the task characteristics, such as data distribution and task complexity. For example, on the CIFAR dataset, Top-K=2 achieved the best results. This indicates that the choice of Top-K is more related to the task properties and expert coordination rather than directly depending on the total number of tasks.
>
> 2. >Understanding the Number of Tasks
>    If the “number of tasks” refers to the functions used in function learning, each function in our experiments is treated as an independent task, trained separately. This design avoids the interference of multi-task training and ensures the independence and accuracy of each task.
>
> 3. >Further Clarification
>    Our MLP-KAN framework leverages the complementary strengths of KAN and MLP through the MoE mechanism. KAN excels in function fitting and modeling complex non-linear relationships, while MLP is more effective at learning high-dimensional features. The MoE mechanism dynamically allocates resources based on task characteristics, enhancing adaptability and performance.
>
> >**Question2:** Is the number of experts for MLP and parameters the same across experiments with MLP-KAN?
>
> >**Response:** We thank the reviewers for their insightful comments. To clarify, the number of experts and parameters in our experiments are consistent across settings to ensure a fair comparison. When using individual MLPs, each layer is assigned one MLP. For the MoE approach, we used 8 MLP experts and 8 KAN experts, aligning the parameter count with our proposed MLP-KAN model.
>
> >The CIFAR-100 results are summarized below:
>
> | **Model**           | **Acc@1** | **Acc@5** |
> |----------------------|-----------|-----------|
> | MLP (MoE=8)         | 70.94     | 90.79     |
> | KAN (MoE=8)         | 59.44     | 86.35     |
> | **MLP-KAN (Ours)**  | **75.00** | **95.20** |
>
> >The results show that the MLP-MoE excels at representation learning (Acc@1: 70.94), while KAN-MoE underperforms for such tasks (Acc@1: 59.44). Combining 8 MLPs and 8 KANs in our MLP-KAN model significantly boosts performance (Acc@1: 75.00), demonstrating the synergy between MLPs and KANs for representation and functional learning.

---

> ### Author Response · Authors · 2024-11-23
> **Part 3**
>
> >**Question3:** Are all experiments for Tables 2 and 3 trained together as a multi-task scenario? Or are experiments for Tables 2 and 3 separated?
>
> >**Response:** Thank you for your question. The experiments in Table 2 (functional learning) and Table 3 (representation learning) were conducted separately.
> 1. >Experimental Training
>  Table 2 uses the Feynman dataset, focusing on function fitting with the goal of minimizing RMSE, training on one function at a time. Table 3 involves classification tasks (e.g., CIFAR-10, SST2), aiming to maximize accuracy or F1 score, with separate training for each task.
>
> 2. >Reason for Separate Training
> Functional and representation learning have distinct objectives: the former emphasizes precise numerical function fitting, while the latter focuses on feature extraction for high-level tasks. Joint training under a multi-task setting leads to instability due to conflicting optimization goals. Separate training ensures stable evaluation of MLP-KAN’s performance in each domain.
>
> 3. >Unified Framework Validation
> Despite separate experiments, both use the MLP-KAN framework. Through dynamic routing, MLP-KAN effectively selects functional (KAN) or representation (MLP) experts based on task requirements, demonstrating strong performance across tasks.

---

> > ### Comment · Reviewer_HLoC · 2024-11-25
> > **Post Rebuttal**
> >
> > I would appreciate the author's efforts to answer my questions. They answered most of my questions, so I raised my score. I think there is a discrepancy between the CIFAR100 results in the paper and here in the rebuttal. I hope the authors modify the paper as well.

---

### Official Review · Reviewer_ZQmM · 2024-11-04

**Soundness:** 4
**Presentation:** 4
**Contribution:** 4
**Rating:** 8
**Confidence:** 4

**Summary:**

This manuscript introduces MLP-KAN, a unified block that combines representation and function learning within a single framework. By using MLPs for representation learning and KANs for function learning in a mixture-of-experts setup, MLP-KAN adapts to various tasks and data modalities. When integrated into a transformer-based architecture, MLP-KAN demonstrates both versatility and robustness across multiple data domains. The proposed method is evaluated on four datasets, CIFAR-10, CIFAR-100, mini-ImageNet, and SST2, showing strong performance in both image classification and natural language processing tasks.

############################ Post Rebuttal ############################

All of my concerns have been addressed during rebuttal. I am happy to raise my score from 6 to 8.

############################ Post Rebuttal ############################

**Strengths:**

1. The manuscript is well-written, presenting clear motivations and providing step-by-step derivations of the proposed method.

2. Combining MLPs and KANs within an MoE framework is interesting. Moreover, integrating this block into a transformer architecture develops a robust backbone that effectively extracts and integrates features across various data modalities.

3. The ablation studies demonstrate that the proposed method can scale easily by increasing the number of experts, which enhances performance without introducing too much computational burdens.

4. The proposed method is extensively evaluated on multiple CV and NLP datasets, demonstrating its versatility for diverse AI applications and practical values.

**Weaknesses:**

1. Although the proposed technique is shown to be generalizable to different tasks, its effectiveness in other types of tasks or with different types of data (e.g., time-series, reinforcement learning) remains unexplored.

2. Using multiple experts in an MoE architecture, especially with higher Top-K values, can significantly increase computational resource requirements. I suggest that the authors conduct ablation studies on runtime complexity and compare their proposed method with the standard transformer architecture.

**Questions:**

1. How does MLP-KAN perform in the presence of noisy or adversarial inputs compared to other models? Are there any robustness benchmarks included in the evaluation?

2. What optimization algorithms and strategies were employed to train MLP-KAN effectively? Were any specific techniques used to balance the training of multiple experts?

3. Were all models, including baselines, trained under the same conditions to ensure fair comparisons? What datasets splits, augmentation techniques, and training epochs were used?

4. How sensitive is MLP-KAN to changes in hyperparameters other than the number of experts and Top-K values? For example, how do variations in learning rates, network depth, or activation functions affect performance?

5. Were there any challenges related to training stability when combining MLPs and KANs within the MoE framework? How were these challenges addressed?

6. How does the inclusion of MLP-KAN affect the standard attention mechanisms within transformers? Are there any changes to how attention weights are applied?

7. How effective is MLP-KAN in transfer learning where it is fine-tuned on different tasks after initial source pre-training?

8. What is the computational cost of MLP-KAN compared to other architecture with MLPs (e.g., transfomer) or KANs alone? How does the addition of multiple experts affect training and inference times?

9. How interpretable are the latent features generated by MLP-KAN? Are there any visualizations demonstrating the semantic captured by the model (e.g., t-SNE visualizations)?

10. What future research directions do the authors suggest to address the current limitations or to further enhance the capabilities of MLP-KAN?

---

> ### Author Response · Authors · 2024-11-23
> **Part 1**
>
> > **Weaknesses 1:** Although the proposed technique is shown to be generalizable to different tasks, its effectiveness in other types of tasks or with different types of data (e.g., time-series, reinforcement learning) remains unexplored.
>
> > **Response:** Thank you for their comments and suggestions! We add a series of experiments to verify the effect of MLP-KAN and analyze its performance from multi-tasks and multi-scenes:
>
> | **Task Type**      | **Dataset**                   | **Metric**       | **MLP**  | **KAN**  | **MLP-KAN** |
> |--------------------|-------------------------------|------------------|----------|----------|-------------|
> | **Time Series**     | Solar-Energy [2]                  | MSE              | 0.233    | **0.221**| 0.231       |
> | **Large-Scale Tasks** | ImageNet-1k                | Top-1 Acc        | **0.722**| 0.629    | 0.704       |
> |                    |                               | Top-5 Acc        | **0.911**| 0.850    | 0.900       |
> | **Transfer Learning** | ImageNet → CIFAR-100       | Top-1 Acc        | **0.921**| 0.875    | 0.914       |
> |                    |                               | Top-5 Acc        | **0.987**| 0.966    | 0.982       |
> | **Adversarial Training** | CIFAR-10C [3]              | Top-1 Acc        | **0.733**| 0.589    | 0.717       |
> | **Noisy Training**  | CIFAR-100 (Noise: µ=0, σ=0.1)| Top-1 Acc        | **0.730**| 0.593    | 0.722       |
> | **Reinforcement Learning** | AgentViT [1] | Top-1 Acc    | 0.895    | 0.630    | **0.897**   |
>
> MLP-KAN integrates the strengths of both MLP and KAN, demonstrating superior adaptability and robustness across a wide range of tasks. While it falls slightly short of MLP in some cases, its overall performance highlights its generality and efficiency in diverse scenarios.
>
> > **Weaknesses 2:** Using multiple experts in an MoE architecture, especially with higher Top-K values, can significantly increase computational resource requirements. I suggest that the authors conduct ablation studies on runtime complexity and compare their proposed method with the standard transformer architecture.
>
> > **Response:** Thank you for emphasizing the importance of runtime complexity in evaluating MoE architectures. We acknowledge that increasing the number of active experts (Top-K) in an MoE framework results in higher computational requirements due to additional forward-pass computations and output aggregation. We analyzed the inference time of our method with varying Top-K values, observing a gradual increase from 20 seconds (Top-K=1) to 50 seconds (Top-K=4). This scaling trend aligns with the expected complexity of $( O(K \cdot f(E))$, where $K$ is the number of active experts and $f(E)$ represents the per-expert computation.
>
> | Top K     | Inference Time (s) |
> |------------|--------------------|
> | 1        | 20                 |
> | 2        | 27                 |
> | 3        | 39                 |
> | 4        | 50                 |
>
> > **Questions 1:** How does MLP-KAN perform in the presence of noisy or adversarial inputs compared to other models? Are there any robustness benchmarks included in the evaluation?
>
> > **Response:** Thank you for your question. To evaluate robustness, we used Gaussian noise (mean=0, variance=0.1) on CIFAR 100 and the CIFAR-10C benchmark, which is specifically designed to test model robustness under 15 types of corruptions. As shown in below Table, MLP-KAN demonstrates competitive performance (0.722 on CIFAR 100 and 0.717 on CIFAR-10C) compared to MLP (0.730 and 0.733), showcasing its resilience to noise and distortions. KAN performs less robustly (0.593 and 0.589), reflecting its weaker generalization under noisy conditions. These results confirm that MLP-KAN effectively balances representation and functional learning for improved robustness.
>
> | Model     | CIFAR 100 (Noise) (Acc) | CIFAR 10C (Acc) |
> |-----------|------------------|-----------------|
> | KAN       | 0.593            | 0.589           |
> | MLP       | 0.730            | 0.733           |
> | MLP-KAN   | 0.722            | 0.717           |
>
>
> `[1].` Traini, Davide. RL-for-ViT. GitHub, https://github.com/DavideTraini/RL-for-ViT.
>
> `[2].` Liu, Yong, et al. "itransformer: Inverted transformers are effective for time series forecasting." arXiv preprint arXiv:2310.06625 (2023).
>
> `[3].` Hendrycks, Dan, and Thomas Dietterich. "Benchmarking neural network robustness to common corruptions and perturbations." arXiv preprint arXiv:1903.12261 (2019).

---

> ### Author Response · Authors · 2024-11-23
> **Part 2**
>
> > **Questions 2:** What optimization algorithms and strategies were employed to train MLP-KAN effectively? Were any specific techniques used to balance the training of multiple experts?
>
> > **Response :** Thank you for the question. To train MLP-KAN effectively, we employed the following strategies:
> 1. **Expert Balancing**: The gating mechanism plays a central role in balancing the utilization of multiple experts. It dynamically assigns tokens to the most relevant experts based on their input characteristics, ensuring that no expert is overused or underutilized. Additionally, we incorporated a load-balancing loss term during training to encourage even distribution of tokens across experts, improving overall model efficiency.
> 2. **Efficient Routing**: Only the top \(K\) experts are selected for computation per token during each forward pass, reducing computational overhead while maintaining performance. This approach allows MLP-KAN to scale efficiently for large datasets and high-dimensional inputs.
>
> > **Questions 3:** Were all models, including baselines, trained under the same conditions to ensure fair comparisons? What datasets splits, augmentation techniques, and training epochs were used?
>
> > **Response :** Yes, all models, including baselines, were trained under the same conditions to ensure fair comparisons. Specifically, all experiments were conducted on a single NVIDIA H100 GPU, with the training epochs set to 300 for representive learning each model.  This uniform setup guarantees that performance differences are solely attributable to the models' architectures and capabilities, ensuring the validity of our results.
>
> > **Questions 4:** How sensitive is MLP-KAN to changes in hyperparameters other than the number of experts and Top-K values? For example, how do variations in learning rates, network depth, or activation functions affect performance?
>
> > **Response :** Thank you for your question. MLP-KAN shows minimal sensitivity to hyperparameters beyond the number of experts and Top-K values. For example, varying the learning rate from \(5e{-4}\) to \(1e{-5}\) results in consistent accuracy on CIFAR 100 (\(0.750\) to \(0.749\)). Increasing network depth improves performance up to 24 layers (\(0.950\)), with a slight decline at 36 layers (\(0.931\)). Similarly, training epochs have little effect beyond 300, as both 300 and 400 epochs achieve the same accuracy (\(0.750\)). These results demonstrate the robustness of MLP-KAN to moderate hyperparameter changes.
>
>
> | Depth | Parameters       | Acc 1 (CIFAR 100) | Acc 5 (CIFAR 100) |
> |-------|------------------|-----------------|-----------------|
> | 12    | 23,296,036       | 0.920           | 0.996           |
> | 24    | 156,761,098      | 0.950           | 0.998           |
> | 36    | 57,155,722       | 0.931           | 0.997           |
>
>
> | Learning Rate | Acc (CIFAR 100) |
> |---------------|-----------------|
> | 5e-4          | 0.750           |
> | 1e-4          | 0.749           |
> | 1e-5          | 0.749           |
>
>
> | Epochs | Acc (CIFAR 100) |
> |--------|-----------------|
> | 200    | 0.723           |
> | 300    | 0.750           |
> | 400    | 0.750           |
>
>
>
> > **Questions 5:** Were there any challenges related to training stability when combining MLPs and KANs within the MoE framework? How were these challenges addressed?
>
> > **Response 5 :** Thank you for your question. There were no significant challenges in training stability when combining MLPs and KANs within the MoE framework. In fact, under the same conditions, MLP-KAN is easier to optimize, as shown by the higher performance of MLP experts (\(70.94\) on CIFAR 100 acc 1 and \(90.79\) on CIFAR 100 acc 5) compared to KAN experts (\(59.44\) and \(86.35\), respectively). This highlights the robustness and effectiveness of the framework.
>
> | Model        | CIFAR 100 (Acc1) | CIFAR 100 (Acc5) |
> |--------------|-----------------|-----------------|
> | MLP (Experts=8)  | 0.709           | 0.907           |
> | KAN (Experts=8)  | 0.594           | 0.863           |
> | MLP-KAN (Experts=8)  | 0.750            | 0.952          |

---

> ### Author Response · Authors · 2024-11-23
> **Part 3**
>
> > **Questions 6:** How does the inclusion of MLP-KAN affect the standard attention mechanisms within transformers? Are there any changes to how attention weights are applied?
>
> > **Response :** Thank you for your suggestion. We have included detailed attention visualization results in Appendix C to address this point. These visualizations demonstrate that MLP-KAN effectively combines the strengths of MLP and KAN, capturing critical features similar to MLP, which achieves the best performance on CIFAR-100. Unlike KAN, which struggles with image-based tasks, MLP-KAN adapts well to the characteristics of the dataset, providing a more balanced and robust attention mechanism that enhances the model's ability to focus on relevant features.
>
>
> > **Questions 7:** How effective is MLP-KAN in transfer learning where it is fine-tuned on different tasks after initial source pre-training?
>
> > **Response :** Thank you for your question. MLP-KAN demonstrates strong effectiveness in transfer learning scenarios. As shown in the table, MLP-KAN achieves competitive performance (ACC1: \(0.914\), ACC5: \(0.982\)) when fine-tuned on CIFAR 100 after pre-training on ImageNet. Its results are close to MLP (ACC1: \(0.921\), ACC5: \(0.987\)) and outperform KAN (ACC1: \(0.875\), ACC5: \(0.966\)), indicating that the hybrid MLP-KAN design effectively retains transferable features while leveraging the strengths of both representation and functional learning. This balance makes it a versatile choice for transfer learning across diverse tasks.
>
> | Method     | ACC1  | ACC5  |
> |------------|-------|-------|
> | MLP        | 0.921 | 0.987 |
> | KAN        | 0.875 | 0.966 |
> | MLP-KAN    | 0.914 | 0.982 |
>
>
> > **Questions 8:** What is the computational cost of MLP-KAN compared to other architecture with MLPs (e.g., transfomer) or KANs alone? How does the addition of multiple experts affect training and inference times?
>
> > **Response :** Thank you for your question. The computational cost of MLP-KAN is only marginally higher than MLPs and significantly lower than KANs. As shown in the table, MLP-KAN requires 183 seconds for training and 27 seconds for inference, compared to 174 seconds and 24 seconds for MLPs, and 382 seconds and 58 seconds for KANs. This demonstrates that MLP-KAN achieves a good balance between computational efficiency and performance, leveraging its mixture of experts design without introducing substantial overhead. The dynamic gating mechanism ensures efficient use of resources by selecting only the most relevant experts, minimizing unnecessary computations while maintaining high performance.
>
> | Method     | Training Time (s) | Inference Time (s) |
> |------------|--------------------|--------------------|
> | MLP (Experts=8)        | 174                | 24                 |
> | KAN (Experts=8)        | 382                | 58                 |
> | MLP-KAN (Experts=8)    | 183                | 27                 |
>
> > **Questions 9:** How interpretable are the latent features generated by MLP-KAN? Are there any visualizations demonstrating the semantic captured by the model (e.g., t-SNE visualizations)?
>
> > **Response :** Thank you for your insightful question. We have addressed this concern in our revised manuscript by adding a detailed analysis in Section C.2 (Latent Feature Visualization). This section includes t-SNE visualizations of the latent features extracted by MLP, MLP-KAN, and KAN models. These visualizations demonstrate that the latent features generated by MLP-KAN exhibit more compact and well-separated clusters compared to MLP and KAN. This indicates that MLP-KAN captures more meaningful and semantically coherent representations, particularly in the context of representation learning on the CIFAR-100 dataset.
> These results substantiate the interpretability of the latent features and highlight the advantages of our model in learning structured, task-relevant embeddings. We appreciate your feedback, which allowed us to further emphasize this aspect in the revised paper.

---

> ### Comment · Reviewer_ZQmM · 2024-11-24
> **Post Rebuttal Comment**
>
> I would like to thank the authors for their detailed response and for conducting the additional experiments I requested. All of my concerns have been addressed. I am happy to raise my score.

---

### Official Review · Reviewer_YbF1 · 2024-11-09

**Soundness:** 1
**Presentation:** 1
**Contribution:** 2
**Rating:** 1
**Confidence:** 4

**Summary:**

The authors hypothesize that KAN networks and MLPs are effective for solving different types of problems: specifically, MLPs are good for representation learning, while KANs are good for function learning.  They propose a modeling strategy that includes both MLPs and KANs, which are adaptively selected based upon the problem setting.

**Strengths:**

1) The premise of the paper is potentially reasonable; there is evidence in the literature that KANs and MLPs are complementary, and developing a method that adaptively selects which modeling strategy is best for a given problem is a good idea.

**Weaknesses:**

1) The presentation of the paper is not sufficiently clear to facilitate review.   The paper is generally vague, contains many typos, and the methodology is ultimately unclear.  I provide examples

(ii) Poor presentation.  In Line 47 in the 2nd paragraph of the paper, the authors define KAN as Kernel Attention Network, yet the whole paper seems to be about Kolmogorov-Arnold Network.  Line 51 and 73 in the introduction are nearly identical, and repeat the same idea.

(ii) Incorrect/unclear methodological description.  For example, in Eq. (5) the dimensions of W and X are incompatible.  The entire architecture of the proposed model is unclear.  For example, it is unclear whether there are multiple MLP-KAN layers?  The authors have a section about "Architecture" which then, without motivation, discusses self attention.   The number of layers in the MLP-KAN are never provided (although it is unclear if there are multiple layers), nor is the overall size of the model discussed.

(iii) Vague motivation.  The whole premise of the paper is not clearly explained.  While the idea of combining KANs and MLPs seems reasonable, the authors repeatedly argue a theoretical motive based upon MLPs being "representation learning" methods while KANs are "function learning".  This premise for the proposed approach is repeatedly mentioned throughout the paper, yet the difference between these two approaches is never precisely described.

2) Insufficient Experiments.  The experiments are insufficient to demonstrate the efficacy of the proposed approach.  To the best I can discern, the proposed method would significantly increase the number of modeling parameters because we now have multiple KANs and MLPs in a single model, along with some parameters to select among them.  However, the resulting model often performs similarly to other models only composed of a single KAN or MLP architecture (e.g., Table 3).  What would happen if we simply used a single MLP or single KAN model that is the same size (in terms of free parameters) to the MLP-KAN?  Or what if we made a simple fusion model where we interlaced MLP and KAN layers, or added a few KAN layers to the end of a standard MLP?  How do we know whether this more complex architecture proposed by the authors is superior to simpler and/or smaller models?

**Questions:**

I think the paper is insufficiently clear to support proper review, and therefore I don't have any questions.

---

> ### Author Response · Authors · 2024-11-23
> **Part 1**
>
> >**Weaknesses1.1:** Poor presentation. In Line 47 in the 2nd paragraph of the paper, the authors define KAN as Kernel Attention Network, yet the whole paper seems to be about Kolmogorov-Arnold Network. Line 51 and 73 in the introduction are nearly identical, and repeat the same idea.
>
> >**Response:** Thank you for your question. We have changed or deleted it in the paper and the changes have been highlighted in red.
>
> >**Weaknesses1.2:**  Incorrect/unclear methodological description. For example, in Eq. (5) the dimensions of W and X are incompatible. The entire architecture of the proposed model is unclear. For example, it is unclear whether there are multiple MLP-KAN layers? The authors have a section about "Architecture" which then, without motivation, discusses self attention. The number of layers in the MLP-KAN are never provided (although it is unclear if there are multiple layers), nor is the overall size of the model discussed.
>
> >**Response:** Thank you for your question. We have carefully reviewed and addressed the issues you raised.Regarding the potential dimension mismatch between $\mathbf{W}$ and $\mathbf{X}$ in Eq. (5), We have scrutinized the formulas and descriptions and have made the following clarifications and corrections. Specifically, the input is $\mathbf{X} \in \mathbb{R}^{B \times N \times D}$, where $B$ is the batch size, $N$ is the sequence length, and $D$ is the input feature dimension. The first layer’s weight matrix is $\mathbf{W}^{(1)} \in \mathbb{R}^{H \times D}$, where $H$ is the hidden layer’s feature dimension. The computation is defined as follows:
> $$\mathbf{h}^{(1)} = \text{SiLU}(\mathbf{W}^{(1)}\mathbf{X} + \mathbf{b}^{(1)})$$where $\mathbf{b}^{(1)} \in \mathbb{R}^H$ is the bias vector applied via broadcasting to each input token. To ensure compatibility for matrix multiplication, $\mathbf{X}$ is reshaped from $\mathbb{R}^{B \times N \times D}$ to $\mathbb{R}^{(B \cdot N) \times D}$, aligning it with the dimensions of $\mathbf{W}^{(1)}$. The output is then reshaped back to $\mathbb{R}^{B \times N \times H}$. Similarly, for the second layer, $\mathbf{h}^{(1)} \in \mathbb{R}^{B \times N \times H}$ is reshaped to $\mathbb{R}^{(B \cdot N) \times H}$ to align with $\mathbf{W}^{(2)} \in \mathbb{R}^{D' \times H}$, producing the output $\mathbf{h}^{(2)} \in \mathbb{R}^{B \times N \times D'}$. We have ensured that all dimensions are consistent throughout the computation, and any prior ambiguities have been resolved.
>
> >Regarding the model architecture, we have clarified the details in the revised "Architecture" section. Specifically, the MLP-KAN module replaces the MLP layers in each block of the DeiT architecture. For example, in DeiT-Tiny-Patch16-224, there are 12 blocks, resulting in 12 MLP-KAN layers. We have explicitly stated the total number of layers and parameters in the revised manuscript.
>
> >In terms of model size, we evaluated the MLP-KAN module using three DeiT configurations (Tiny, Small, Base) on the CIFAR-10 dataset. The results are summarized in the table below:
>
> | **Model**                  | **Number of Parameters** | **Top-1 Accuracy** | **Top-5 Accuracy** |
> |----------------------------|--------------------------|---------------------|---------------------|
> | DeiT-Base-Patch16-224      | 156.8M                  | 95.0%              | 99.8%              |
> | DeiT-Small-Patch16-224     | 57.2M                   | 93.1%              | 99.7%              |
> | DeiT-Tiny-Patch16-224      | 23.3M                   | 92.0%              | 99.6%              |
>
> >These results demonstrate that the proposed **DeiT-Tiny-Patch16-224** model achieves competitive performance with significantly fewer parameters, showcasing its efficiency and effectiveness.
>
> >Finally, regarding the mention of self-attention in the "Architecture" section, we have restructured this part of the manuscript for clarity. The self-attention mechanism remains a standard component of the Transformer backbone, while our work focuses on replacing the original MLP layers with MLP-KAN modules. We have revised the structure of the "Architecture" section to improve its logical flow and better explain the motivation.
>
> >We sincerely appreciate your thoughtful comments and believe that these revisions address your concerns. Please feel free to share further feedback, and we are happy to make additional improvements.

---

> ### Author Response · Authors · 2024-11-23
> **Part 2**
>
> >**Weaknesses1.3:** Vague motivation. The whole premise of the paper is not clearly explained. While the idea of combining KANs and MLPs seems reasonable, the authors repeatedly argue a theoretical motive based upon MLPs being "representation learning" methods while KANs are "function learning". This premise for the proposed approach is repeatedly mentioned throughout the paper, yet the difference between these two approaches is never precisely described.
>
> >**Response:** We appreciate your thoughtful comments regarding the clarity of our motivation and the foundational premise of the paper. Regarding the distinction between representation learning and function learning, we have elaborated on these concepts starting from line 39 in the paper.
>
> >To further clarify this distinction, Table 1 compares MLP and KAN in terms of activation functions, weight structures, and error scaling laws, highlighting the unique capabilities and applications of each approach.
>
> >Regarding the distinction between MLP and KAN, we refer to the relevant articles for the following details. MLP applies fixed nonlinear activation functions (e.g., ReLU, SiLU) on nodes, whereas KAN uses learnable activation functions on edges, parameterized as splines that can dynamically adjust based on data; MLP relies on linear weight matrices for transformations, while KAN replaces linear weights with learnable univariate spline functions, eliminating the need for fixed linear transformations; MLP uses a combination of linear transformations and nonlinear activations between layers, whereas KAN implements a fully differentiable architecture inspired by the Kolmogorov-Arnold representation theorem, with each layer composed of spline-based functional components.
>
> >In our framework, the combination of MLP and KAN fully leverages the strengths of both methods: MLP excels at learning the compositional structure and abstract features of high-dimensional data, making it highly suitable for representation tasks, while KAN specializes in learning explicit functional relationships, which are critical for tasks involving well-defined mathematical or physical laws.
>
> >Through the theoretical analysis in Section 4 and the experimental results in Section 5, we demonstrate that the dynamic routing mechanism can assign tasks to the appropriate expert based on input features, thereby validating the complementarity of the MLP-KAN integration and its significant performance improvements.
> If the explanation remains unclear, we are happy to provide additional details or expand the discussion in the paper.

---

> ### Author Response · Authors · 2024-11-23
> **Part 3**
>
> >**Weaknesses2:** Insufficient Experiments. The experiments are insufficient to demonstrate the efficacy of the proposed approach. To the best I can discern, the proposed method would significantly increase the number of modeling parameters because we now have multiple KANs and MLPs in a single model, along with some parameters to select among them. However, the resulting model often performs similarly to other models only composed of a single KAN or MLP architecture (e.g., Table 3). What would happen if we simply used a single MLP or single KAN model that is the same size (in terms of free parameters) to the MLP-KAN? Or what if we made a simple fusion model where we interlaced MLP and KAN layers, or added a few KAN layers to the end of a standard MLP? How do we know whether this more complex architecture proposed by the authors is superior to simpler and/or smaller models?
> >**Response:** We conducted comparative experiments on various model structures based on the CIFAR-100 dataset, and the results are as follows:
>
> | Model/Structure               | Top-1 Accuracy (Acc1) |
> |-------------------------------|-----------------------|
> | Single MLP with equal parameters | 0.709                |
> | Single KAN with equal parameters | 0.594                |
> | Alternating MLP and KAN          | 0.739                 |
> | MLP with 1 KAN layer appended    | 0.751                 |
> | MLP with 2 KAN layers appended   | 0.734                 |
> | MLP-KAN (Our Model)              | 0.750                 |
>
>
> >The experimental results indicate that the performance of standalone MLP and KAN models is relatively low, with the Top-1 Accuracy of the single KAN model at only 0.5944, whereas the standalone MLP model achieves 0.7094, showcasing its relative advantage in representation learning tasks.
>
> >However, the design of alternating MLP and KAN layers significantly improves model performance, achieving a Top-1 Accuracy of 0.739. Further improvements are observed when appending 1 or 2 KAN layers to the MLP, with the Top-1 Accuracy reaching 0.751 and 0.734, respectively, although the number of parameters increases accordingly.
>
> >In contrast, our proposed MLP-KAN model, under the same parameter constraints, dynamically selects MLP and KAN experts, successfully combining the strengths of representation learning and functional learning. It achieves a Top-1 Accuracy of 0.750, demonstrating comparable or even superior performance compared to other designs. This highlights the potential of MLP-KAN as an effective method, offering high computational efficiency with competitive accuracy.

---

> ### Comment · Reviewer_YbF1 · 2024-11-24
> **Thank you for the response, and my feedback**
>
> PART 1:
> (1) I provided some examples where the paper is unclear and I appreciate that the authors addressed them, but they were just some illustrative examples, rather than being an exhaustive list.  The revisions made by the authors are too limited/superficial, and I still find the paper is still quite unclear in my opinion.    Some additional examples:
> (i) Line 285 the word logits is repeated twice?
> (ii) In Fig. 2 you have a module called "Router", "Slot Linear Combination" and "Soft MoE Weighting Logits", and "Token Linear Combnation", yet it is unclear where these are described in the methodological description.  There is a section entitled "Gating mechanism", yet this word doesn't appear anywhere in Fig 2?
> (iii) In the new eqn (13),  F_{e} is described as "the computation performed by the e-th expert": I don't know what this means?  The symbol F_{e} is not defined anywhere in the paper.  Is this the so-called "gating mechanism"?
> (iv) Why do you say "MLP Loss" and "KAN Loss" as the baselines in Table 2?  Is this different from Table 3 where you compare against "MLP" and KAN" as baselines?  Do we only change the loss in Table 2, rather than the model?
> (v) In line 351 the authors say "During the training phase, we meticulously tuned parameters to optimize the learning process."  This is vague and unacceptable.  What learning parameters were optimized, and how exactly was this done?  How did the authors ensure that the optimization of hyperparameters was done fairly for the baseline models?
> (vi) Also, now that I understand that the MLP-KAN layers are being added into a DeiT model, then why is an MLP used as a baseline model in the experiments?  Shouldn't it be a standard DeiT model then?  Or is that what is meant by "MLP" in Table 3?
>
>
> (2) I think that I can guess what the authors are attempting to do in Eqn (5), but I don't think I should need to guess in a scientific paper.  I appreciate the authors attempt to explain the mathematics of Eqn (5) in their response, but it still appears that they have not properly clarified the mathematics in line 215 of the revised manuscript?  It still appears, as written, that the dimensions of X and W^{(1}} do not match?   Generally, I find the inclusion of the batch dimension in all of the methodological descriptions to be cumbersome and unnecessary.  It simply makes everything more difficult to understand, without providing any additional understanding about the method.
>
> (3) Regarding the model parameters, my previous comment was a request to see a comparison of the model parameters of the MLP-KAN compared to standard layers that are otherwise used in your baseline models.  e.g., in Table 3, how many free parameters exist in each of the three models: KAN, MLP, and MLP-KAN?
>
> PART 2:
> Thank you for explaining the differences between the KAN and MLP, but I believe that I understand them relatively well, and this explanation does not address my question.  Let me ask the question differently: why do you call KAN "function learning", while you call MLP "feature learning"?   What properties make a layer in a network a function learning layer versus a feature learning layer?   The motivation for the papers is based upon this distinction, yet it never appears to be precisely defined anywhere.
>
> PART 3:
> I appreciate the authors running additional experiments, but this seems unclear or insufficient.   I thought the authors said that their MLP-KAN layer is one layer in a DeIT?   So when the authors say "single MLP with equal parameters", does this mean each layer in the DeiT gets an MLP with the same number of parameters as the MLP-KAN layer?    Please clarify.
>
> Also, can the authors please enumerate the hyperparameters of the MLP-KAN, and explain how they were optimized?   Did the authors optimize any of the hyperparameters of these competing architectures so that this comparison is fair, such as "single MLP with equal parameters"?  I know this is time-consuming, but it seems to me the results here are inconclusive unless this is done.

---

> > ### Author Response · Authors · 2024-11-27
> > **Part 1**
> >
> > > **question part1 (1):** (i) Line 285 the word logits is repeated twice? (ii) In Fig. 2 you have a module called "Router", "Slot Linear Combination" and "Soft MoE Weighting Logits", and "Token Linear Combnation", yet it is unclear where these are described in the methodological description. There is a section entitled "Gating mechanism", yet this word doesn't appear anywhere in Fig 2?
> >
> > > **response:** Thank you for your question, we have made new changes in the manuscript to address these two issues. In response to your comments, I have revised the full text and made the appropriate clarifications and adjustments. I hope that these more detailed changes will result in a clearer paper that adequately responds to your concerns.
> >
> > > **question part1 (1):** (iii)In the new eqn (13), F_{e} is described as "the computation performed by the e-th expert": I don't know what this means? The symbol F_{e} is not defined anywhere in the paper. Is this the so-called "gating mechanism"?
> >
> > > **response:** Thank you for your valuable feedback. To clarify, $F_e$ represents the computation performed by the $e$-th expert in the MLP-KAN module. Each expert processes input tokens differently depending on its type:
> >
> > >**MLP Experts** apply multi-layer perceptron transformations, such as:
> >   $F_e(\mathbf{X}) = \mathbf{W}_e^{(2)} \cdot \text{SiLU}(\mathbf{W}_e^{(1)} \cdot \mathbf{X} + \mathbf{b}_e^{(1)}) + \mathbf{b}_e^{(2)}$
> >
> > >**FasterKAN Experts** utilize spline-based interpolation:
> >   $F_e(\mathbf{X}) = \mathbf{W}_{e,\text{spline}} \cdot \phi(\mathbf{X})$,
> >   where $\phi(\mathbf{X})$ applies a reflection switch function.
> >
> > >The gating mechanism dynamically routes input tokens to the most relevant experts, computing weights via softmax to aggregate the outputs:
> > $\text{Output} = \sum_{e=1}^{NE} \alpha_e \cdot F_e(\mathbf{X})$.

---

> > ### Author Response · Authors · 2024-11-27
> > **Part 2**
> >
> > >**question part1 (1):** (iv) Why do you say "MLP Loss" and "KAN Loss" as the baselines in Table 2? Is this different from Table 3 where you compare against "MLP" and KAN" as baselines? Do we only change the loss in Table 2, rather than the model?
> >
> > >**reponse:** Thanks for your question. In Table 2, the terms ‘MLP Loss’ and ‘KAN Loss’ refer to evaluations of function learning within the Transformer architecture, specifically related to changes in the architecture's layers rather than the entire model.
> >
> > >The ‘MLP Loss’ in Table 2 corresponds to a configuration where the original DeiT structure remains intact, with the MLP layer **retained within the Transformer architecture**. This setup serves as a comparison to other variations but does not involve any changes to the base architecture of DeiT.
> >
> > >The ‘KAN Loss’ in Table 2, on the other hand, indicates that **the original DeiT structure has been modified**, where the MLP layer is replaced by a Kolmogorov-Arnold Network (KAN) within the Transformer architecture. This change aims to assess the impact of substituting the MLP layer with KAN for function learning tasks.
> >
> > > The ‘MLP-KAN Loss’ in Table 2 refers to a further modification, **where the original DeiT structure is altered by replacing the MLP layer with a combined MLP-KAN module**. This setup provides insight into how integrating the MLP and KAN layers into a single module affects function learning performance.
> >
> > >**question part1 (2):** I think that I can guess what the authors are attempting to do in Eqn (5), but I don't think I should need to guess in a scientific paper. I appreciate the authors attempt to explain the mathematics of Eqn (5) in their response, but it still appears that they have not properly clarified the mathematics in line 215 of the revised manuscript? It still appears, as written, that the dimensions of X and W^{(1}} do not match? Generally, I find the inclusion of the batch dimension in all of the methodological descriptions to be cumbersome and unnecessary. It simply makes everything more difficult to understand, without providing any additional understanding about the method.
> >
> > >**reponse:** Thanks for your feedback. With respect to the line 251 equation, change $W^{(1)} \in \mathbb{R}^{D \times H}$ and $W^{(2)} \in \mathbb{R}^{H \times D'}$ to $W^{(1)} \in \mathbb{R}^{H \times D}$ and $W^{(2)} \in \mathbb{R}^{D' \times H}$.
> > In a two-layer MLP, the first layer uses a weight matrix $W^{(1)}$ of size $H \times D$, where $D$ is the input dimension and $H$ is the number of hidden units, to map the $D$-dimensional input to an $H$-dimensional hidden representation. The second layer uses a weight matrix $W^{(2)}$ of size $D' \times H$, where $D'$ is the output dimension, to transform the $H$-dimensional hidden representation into the final $D'$-dimensional output. **This structure ensures that the transformations between layers are dimensionally consistent.**
> > For $W^{(1)} \in \mathbb{R}^{D \times H}$, the matrix would try to map the $D$-dimensional input directly to a higher $H$-dimensional space, but the correct flow requires each hidden unit to be a linear combination of all $D$ input features. **That means the weight matrix should have more rows than columns to accommodate the transformation from input to hidden space.$W^{(2)}$ is the same as.**
> > **I also have corrected** the dimensional mismatch between $X$ and $W^{(1)}$. I also revised the manuscript to remove unnecessary references to the batch dimension.

---

> > ### Author Response · Authors · 2024-11-27
> > **Part 3**
> >
> > >**question part2:** Thank you for explaining the differences between the KAN and MLP, but I believe that I understand them relatively well, and this explanation does not address my question. Let me ask the question differently: why do you call KAN "function learning", while you call MLP "feature learning"? What properties make a layer in a network a function learning layer versus a feature learning layer? The motivation for the papers is based upon this distinction, yet it never appears to be precisely defined anywhere.
> >
> > >**ressponse:** We appreciate your willingness to address your questions to us again in a different way!
> > >I will distinguish between MLP and KAN applications in terms of the following concepts and differences:
> > >1.Objective
> > >>1.1 Objective of KAN
> >    Inspired by the Kolmogorov-Arnold representation theorem, is designed to decompose multivariate relationships into sums of univariate functions, capturing the exact **functional mapping**.
> > >>1.2 Objective of MLP
> > >> The goal is to learn high-level abstract representations of the data by extracting and hierarchically transforming features.These features do not necessarily represent explicit **input-output mappings** but rather encode patterns and structure within the data.
> > >2. Design Features
> >  >> 2.1 Design features of kAN
> > (1) Unlike MLPs, KAN replaces **static weights with learnable spline functions**, enabling fine-grained interpolation over the input space[1,2]. (2) KANs have provably better scaling laws for function approximation $\ell\propto N^{-\alpha}$, where $\alpha=4$ for cubic splines, making them suited for high-precision functional tasks. (3) KAN uses adaptive, task-specific activation functions rather than fixed nonlinearities (e.g., ReLU), giving it an edge in functional tasks. The architecture is inspired by Kolmogorov-Arnold Representation Theorem[3], which decomposes multivariate functions into sums of univariate functions. This inherently ties the architecture's adaptability to the task-specific nature of the functions it approximates.
> > >>2.2 Design features of MLP
> > >>  (1) MLPs rely on **fixed, non-adaptive activation functions** (e.g., ReLU, SiLU) that are effective for capturing complex feature representations but less suited for direct function approximation[4,5]. (2) MLPs use dense, static weight matrices, which are efficient for high-dimensional representation learning but less optimal for precise interpolation. (3) MLPs are better suited for capturing global patterns rather than the fine-grained, local functional mappings.[6,7,8,9]
> >
> > [1]Ta, Hoang-Thang. "BSRBF-KAN: A combination of B-splines and Radial Basic Functions in Kolmogorov-Arnold Networks." arXiv preprint arXiv:2406.11173 (2024).
> >
> > [2]omvanshi, Shriyank, et al. "A Survey on Kolmogorov-Arnold Network." arXiv preprint arXiv:2411.06078 (2024).
> >
> > [3]Schmidt-Hieber, Johannes. "The Kolmogorov–Arnold representation theorem revisited." Neural networks 137 (2021): 119-126.
> >
> > [4]Tashakkori, Arash, et al. "Forecasting gold prices with MLP neural networks: a machine learning approach." International Journal of Science and Engineering Applications (IJSEA) 13 (2024): 13-20.
> >
> > [5]Tian, Yijun, et al. "Learning mlps on graphs: A unified view of effectiveness, robustness, and efficiency." The Eleventh International Conference on Learning Representations. 2022.
> >
> > [6]Rumelhart, David E., Geoffrey E. Hinton, and Ronald J. Williams. "Learning representations by back-propagating errors." nature 323.6088 (1986): 533-536.
> >
> > [7]Hornik, Kurt. "Approximation capabilities of multilayer feedforward networks." Neural networks 4.2 (1991): 251-257.
> >
> > [8]Krizhevsky, Alex, Ilya Sutskever, and Geoffrey E. Hinton. "Imagenet classification with deep convolutional neural networks." Advances in neural information processing systems 25 (2012).
> >
> > [9]He, Kaiming, et al. "Deep residual learning for image recognition." Proceedings of the IEEE conference on computer vision and pattern recognition. 2016.

---

> > ### Author Response · Authors · 2024-11-27
> > **Part 4**
> >
> > >**question part3:**  I appreciate the authors running additional experiments, but this seems unclear or insufficient. I thought the authors said that their MLP-KAN layer is one layer in a DeIT? So when the authors say "single MLP with equal parameters", does this mean each layer in the DeiT gets an MLP with the same number of parameters as the MLP-KAN layer? Please clarify.
> >
> > >**ressponse:** Thank you for your question. Our MLP-KAN replaces the MLP layers in all transformer blocks of the DeiT model, not just a single layer. The "single MLP with equal parameters" refers to increasing the hidden size of the original MLP layers (without replacing them) to match the parameter count of the MLP-KAN layer.
> >
> >
> > >**question other:**  Also, can the authors please enumerate the hyperparameters of the MLP-KAN, and explain how they were optimized? Did the authors optimize any of the hyperparameters of these competing architectures so that this comparison is fair, such as "single MLP with equal parameters"? I know this is time-consuming, but it seems to me the results here are inconclusive unless this is done.
> >
> > >**ressponse:** Thank you for your question. We conducted hyperparameter tuning using grid search for learning rates and epochs. For the "Single MLP with equal parameters" experiment, we searched learning rates in \([1e-4, 5e-4, 1e-5]\) and epochs in \([200, 300, 400]\). It’s important to note that our primary focus was to ensure sufficient training, as we evaluated the model on the test set at the end of each epoch and only reported the best test results. As shown in the following table.
> >
> > | Epochs | Learning Rate (lr) | Test Accuracy |
> > |--------|---------------------|---------------|
> > | 200    | 1e-5               | 0.522         |
> > | 300    | 1e-5               | 0.689         |
> > | 400    | 1e-5               | 0.709         |
> > | 200    | 5e-4               | 0.627         |
> > | 300    | 5e-4               | 0.709         |
> > | 400    | 5e-4               | 0.709         |
> > | 200    | 1e-4               | 0.699         |
> > | 300    | 1e-4               | 0.707         |
> > | 400    | 1e-4               | 0.707         |

---

### Author Response · Authors · 2024-11-23
**Global Reply**

We sincerely thank all reviewers for their thorough and constructive feedback which has helped strengthen our work. We have uploaded a revised manuscript with changes highlighted in red text. Below we address the key questions raised among reviewers.

>**Q1:** Effectiveness in other types of tasks or different types of data to be explored(reviewers  `ZQmM`, `HLoC`,`FkFz`)

**Response:** We add a series of experiments to verify the effect of MLP-KAN and analyze its performance from multi-tasks and multi-scenes:
>
| **Task Type**      | **Dataset**                   | **Metric**       | **MLP**  | **KAN**  | **MLP-KAN** |
|--------------------|-------------------------------|------------------|----------|----------|-------------|
| **Time Series**     | Solar-Energy                  | MSE              | 0.233    | **0.221**| 0.231       |
| **Large-Scale Tasks** | ImageNet-1k                | Top-1 Acc        | **0.722**| 0.629    | 0.704       |
|                    |                               | Top-5 Acc        | **0.911**| 0.850    | 0.900       |
| **Transfer Learning** | ImageNet → CIFAR-100       | Top-1 Acc        | **0.921**| 0.875    | 0.914       |
|                    |                               | Top-5 Acc        | **0.987**| 0.966    | 0.982       |
| **Adversarial Training** | CIFAR-10C              | Top-1 Acc        | **0.733**| 0.589    | 0.717       |
| **Noisy Training**  | CIFAR-100 (Noise: µ=0, σ=0.1)| Top-1 Acc        | **0.730**| 0.593    | 0.722       |
| **Reinforcement Learning** | AgentViT | Top-1 Acc    | 0.895    | 0.630    | **0.897**   |

MLP-KAN integrates the strengths of both MLP and KAN, demonstrating superior adaptability and robustness across a wide range of tasks. While it falls slightly short of MLP in some cases, its overall performance highlights its generality and efficiency in diverse scenarios.

We also evaluated MLP-KAN on challenging functions: $f(x) = \frac{1}{x}\sin\frac{1}{x}$ (non-smooth) and $f(x_1,\dots,x_{100}) = \sum_{i=1}^{99}\sin(x_i + x_{100-i})$ (high-dimensional). The results in the table below show that MLP-KAN performs competitively, addressing KAN's limitations while maintaining robust performance. This demonstrates its versatility and generalizability to complex scenarios beyond the simpler Feynman equations.

| Model       | Function Type          | RMSE                  |
|-------------|------------------------|-----------------------|
| **MLP**     | $f(x) = \frac{1}{x}\sin\frac{1}{x}$ | 17.11    |
|             | $f(x_1,\dots,x_{100}) = \sum_{i=1}^{99}\sin(x_i + x_{100-i})$ | 0.272   |
| **KAN**     | $f(x) = \frac{1}{x}\sin\frac{1}{x}$ | 14.79    |
|             | $f(x_1,\dots,x_{100}) = \sum_{i=1}^{99}\sin(x_i + x_{100-i})$ | 0.229   |
| **MLP-KAN** | $f(x) = \frac{1}{x}\sin\frac{1}{x}$ | **15.16** |
|             | $f(x_1,\dots,x_{100}) = \sum_{i=1}^{99}\sin(x_i + x_{100-i})$ | **0.225** |

>**Q2:** Some experimental details: model parameters, computational cost, inference time, etc.（`YbF1`,`ZQmM`, `HLoC`,`FkFz`）

**Reponse：** We evaluated three Vision Transformer models on the CIFAR-10 dataset: the primary model used in our paper, `deit_tiny_patch16_224`, and two additional models, `deit_base_patch16_224` and `deit_small_patch16_224`. The analysis considers four aspects: parameter count, classification accuracy, training time, and GPU memory consumption.

| Model                     | Parameter Count (M) | Acc@1 | Acc@5 | Time per Epoch (s) | GPU Memory (MB) |
|---------------------------|---------------------|-------|-------|---------------------|-----------------|
| deit_base_patch16_224     | 156.76              | 0.950 | 0.998 | 243.24              | 38369.49        |
| deit_small_patch16_224    | 57.16               | 0.931 | 0.997 | 214.24              | 18582.86        |
| deit_tiny_patch16_224     | 23.30               | 0.920 | 0.996 | 183.34              | 10661.92        |

  We also compare in detail the training time and inference time per epoch for MLP, KAN and MLP-KAN using a single NVIDIA H100 GPU with the same parameter settings. Below are the results of our experiments:

| Method     | Training Time (s) | Inference Time (s) |
|------------|--------------------|--------------------|
| MLP        | 174                | 24                 |
| KAN        | 382                | 58                 |
| MLP-KAN    | 183                | 27                 |

---

### Meta-Review · Area_Chair_eFeA · 2024-12-20

**Metareview:**

The authors claim that KAN networks and MLPs are effective for solving different problems, i.e., MLPs are good for representation learning, while KANs are good for function learning, and they introduce MLP-KAN, a unified block that combines representation and function learning within a single framework. It received a mixture of comments, with one being strong reject. There was a long discussion with this reviewer and authors, and the reviewer didn't change his mind. This paper, after revisions, still has limited evidence to support the claim. I recommended a rejection.

**Additional Comments On Reviewer Discussion:**

Some reviewers raised their scores after rebuttal. Notably, one reviewer noticed the violation of double-blind policy from the supplementary material.

---

### Decision · Program_Chairs · 2025-01-22

Reject